# Topological constraints in early multicellularity favor reproductive division of labor

**David Yanni[1][‡], Shane Jacobeen[1][‡], Pedro Márquez-Zacarías[2,3], Joshua S Weitz[1,3], William C Ratcliff[3][†]*, Peter J Yunker[1][†]***

[1]School of Physics, Georgia Institute of Technology, Atlanta, United States; [2]Interdisciplinary Graduate Program in Quantitative Biosciences, Georgia Institute of Technology, Atlanta, United States; [3]School of Biological Sciences, Georgia Institute of Technology, Atlanta, United States

**Abstract** Reproductive division of labor (e.g. germ-soma specialization) is a hallmark of the evolution of multicellularity, signifying the emergence of a new type of individual and facilitating the evolution of increased organismal complexity. A large body of work from evolutionary biology, economics, and ecology has shown that specialization is beneficial when further division of labor produces an accelerating increase in absolute productivity (i.e. productivity is a convex function of specialization). Here we show that reproductive specialization is qualitatively different from classical models of resource sharing, and can evolve even when the benefits of specialization are saturating (i.e. productivity is a concave function of specialization). Through analytical theory and evolutionary individual-based simulations, we demonstrate that reproductive specialization is strongly favored in sparse networks of cellular interactions that reflect the morphology of early, simple multicellular organisms, highlighting the importance of restricted social interactions in the evolution of reproductive specialization.

**\*For correspondence:**
william.ratcliff@gatech.edu (WCR);
peter.yunker@gatech.edu (PJY)

[†]These authors contributed equally to this work
[‡]These authors also contributed equally to this work

**Competing interests:** The authors declare that no competing interests exist.

## Introduction

The evolution of multicellularity set the stage for unprecedented increases in organismal complexity (*Szathmáry and Smith, 1995*; *Knoll, 2011*). A key factor in the remarkable success of multicellular strategies is the ability to take advantage of within-organism specialization through cellular differentiation (*Queller and Strassmann, 2009*; *Brunet and King, 2017*; *Cavalier-Smith, 2017*). Reproductive specialization, which includes both the creation of a specialized germ line during ontogeny (as in animals and volvocine green algae) and functional differentiation into reproductive and non-reproductive tissues (as in plants, green and red macroalgae, and fungi), may be especially important (*Cooper and West, 2018*; *Michod et al., 2006*; *Ispolatov et al., 2012*; *Solari et al., 2013*; *Michod, 2007*; *West et al., 2015*). Reproductive specialization is an unambiguous indication that biological individuality rests firmly at the level of the multicellular organism (*Michod, 1999*; *Folse and Roughgarden, 2010*), and is thought to play an important role in spurring the evolution of further complexity by inhibiting within-organism (cell-level) evolution (*Buss, 1988*) and limiting reversion to unicellularity (*Libby and Ratcliff, 2014*). Despite the central importance of reproductive specialization, its origin and further evolution during the transition to multicellularity remain poorly understood (*McShea, 2000*).

The origin of specialization has long been of interest to evolutionary biologists, ecologists, and economists. A large body of theory from these fields shows that specialization pays off only when it increases total productivity, compared to the case where each individual simply produces what they need (*Szathmáry and Smith, 1995*; *Smith and Szathmáry, 1997*; *Goldsby et al., 2012*;

*Corning and Szathmáry, 2015*; *Hidalgo and Hausmann, 2009*; *Boza et al., 2014*; *Taborsky et al., 2016*; *Page et al., 2006*; *Rueffler et al., 2012*; *Szekely et al., 2013*; *Findlay, 2008*; *Amado et al., 2018*). Certain types of trading arrangements maximize the benefits of specialization; highly reciprocal interactions, which facilitate exchange between complementary specialists, amplify cooperation (*Allen et al., 2017*; *Pavlogiannis et al., 2018*). Still, previous work finds that even when groups grow in an ideal spatial arrangement, increased specialization and trade is only favored by natural selection when productivity increases as an accelerating function of the degree of specialization (i.e., productivity is a convex, or super-linear, function of the degree of specialization). Conversely, saturating functional returns (i.e. productivity is a concave, or sub-linear, function of the degree of specialization) should inhibit the evolution of specialization (*Cooper and West, 2018*; *Michod et al., 2006*; *Ispolatov et al., 2012*; *Solari et al., 2013*; *Michod, 2007*; *West et al., 2015*).

Reproductive specialization differs from classical models of trade in several key respects. Trade between germ (reproductive) and somatic (non-reproductive) cells is intrinsically asymmetric, because the cooperative action, multicellular replication, is not a product that is shared evenly. Selection acts primarily on the fitness of the multicellular group as a whole (*Folse and Roughgarden, 2010*). As a result, optimal specialization can result in behaviors that reduce the short-term fitness of some cells within the multicellular group (*Michod et al., 2006*; *Michod, 2007*), often manifest as reproductive altruism.

Understanding the evolution of cell-cell trade, a classic form of social evolution (*Kirk, 2005*), requires understanding the extent of between-cell interactions. Network theory has proven to be an exceptionally powerful and versatile technique for analyzing social dynamics (*Wey et al., 2008*; *Lieberman et al., 2005*), and indeed, is uniquely well suited to understanding the evolution of early multicellular organisms. When cells adhere through permanent bonds, sparse network-like bodies (i.e. filaments and trees) often result (*Amado et al., 2018*). This mode of group formation is not only common today among simple multicellular organisms (*Umen, 2014*; *Claessen et al., 2014*), but is the dominant mode of group formation in the lineages evolving complex multicellularity (i.e. plants, red algae, brown algae, and fungi, but not animals).

In this paper, we develop and investigate a model for how the network topology of early multicellular organisms affects the evolution of reproductive specialization. We find that under a broad class of sparse networks, complete functional specialization can be adaptive even when returns from dividing labor are saturating (i.e. concave/sub linear). Sparse networks impose constraints on who can share with whom, which counterintuitively increases the benefit of specialization (*McShea, 2000*). By dividing labor, multicellular groups can capitalize on high between-cell variance in behavior, ultimately increasing group-level reproduction. Further, we consider group morphologies that naturally arise from simple biophysical mechanisms and show that these morphologies strongly promote reproductive specialization. Our results show that reproductive specialization can evolve under a far broader set of conditions than previously thought, lowering a key barrier to major evolutionary transitions.

## Model

Reproductive specialization can be modeled as the separation of two key fitness parameters, those related to either viability or fecundity, into separate cells within the multicellular organism (*Michod, 2006*; *Folse and Roughgarden, 2010*). The dichotomy of viability versus fecundity was originally used by *Michod, 2006* to partition components of cellular fitness into actions that contribute to keeping a cell alive (viability), and actions that directly contribute to reproduction (fecundity). Multicellular organisms often have evolved to divide labor along these two lines (i.e. reproduction by germ cells and survival provided by somatic cells), while their unicellular ancestors had to do both. We define viability as activities keeping the cell alive (e.g. investing in cellular homeostasis or behaviors that improve survival), and fecundity as activities involved in cellular reproduction.

At the cellular level, there appears to be a fundamental asymmetry in how viability effort and fecundity effort can be shared among cells: while multicellular organisms readily evolve differentiated cells that are completely reliant on helper cells (i.e. glial cells that support neurons in animals or companion cells that support sieve tube cells in plants), no cell can directly share its ability to reproduce. To better understand the intuition behind this, consider a cell that elongates prior to fission. This cell must grow to approximately twice its original length. Two cells cannot elongate by 50% and then combine their efforts; elongation is an intrinsically single cell effort. We thus use a model in

which viability can be shared across connected cells, but fecundity cannot be shared (note, in order to test the sensitivity of our predictions to this assumption, in a later section we will consider the more general case in which viability and fecundity can both be shared, but by different amounts).

We consider a model of multicellular groups composed of clonal cells that each invest resources into viability and fecundity. Because there is no within-group genetic variation, within-group evolution is not possible, though selection can act on group-level fitness differences. Specifically, we consider the pattern of cellular investment in fecundity and viability, and their sharing of these resources with neighboring cells within the group, to be the result of a heritable developmental program. Thus, selection is able to act on the multicellular fitness consequences of different patterns of cellular behavior within the group. We let $v$ denote each cell's investment into viability, and $b$ denote each cell's investment into fecundity. Each cell's total investment is constrained so that $v + b = 1$. However, a cell's return on its investment is in general nonlinear. Here, we let $\alpha$ represent the 'return on investment exponent': by tuning $\alpha$ above and below 1.0, we can simulate conditions with accelerating and saturating (i.e. convex and concave, or super- and sub-linear) returns on investment, respectively. We let $\tilde{v}$ and $\tilde{b}$ represent a cell's return on viability and fecundity investments, respectively. Following **Michod, 2006**; **Michod and Roze, 1997**, we calculate a cell's reproductive output as a multiplicative function of $\tilde{v}$ and $\tilde{b}$ (thus, both functions must be positive for a cell to grow). A single cell's reproduction rate is $w = \tilde{v}\tilde{b} = v^\alpha b^\alpha$. At the group level, fitness is the total contribution of all cells in the group toward the production of new groups (i.e. group level reproduction). The group level fitness is thus the sum of $\tilde{v}\tilde{b}$ over all cells.

Finally, cells may share the products of their investment in viability with other cells to whom they are connected. For a given group, the details about who may share with whom, and how much, is encoded in a weighted adjacency matrix $\mathbf{c}$. The element $c_{ij}$ defines what proportion of viability returns cell $i$ shares with cell $j$. Cells cannot give away all of their viability returns, as they would no longer be viable; mathematically, we count a cell among its neighbors and thus ensure that they always 'share' a positive portion of viability returns with themselves, so that $c_{ii} > 0$. Furthermore, since a cell cannot share more viability returns than the total it possesses, we have $\sum_{i=1}^{N} c_{ji} = 1$ for a group of $N$ cells. For the networks we consider, each cell takes a fraction $\beta$ of its viability returns and shares that fraction equally among all of its $n_i$ neighbors (including itself), and keeps the rest of its returns $1 - \beta$ for itself. Therefore cell $i$ keeps a total fraction of $1 - \beta + \frac{\beta}{n_i}$ of its returns for itself and gives $\frac{\beta}{n_i}$ to each of its non-self neighbors. In other words, $c_{ii} = 1 - \beta + \frac{\beta}{n_i}$, $c_{ij} = \frac{\beta}{n_i}$ if cells $i$ and $j$ are connected, and $c_{ij} = 0$ if cells $i$ and $j$ are not connected. This means the total amount of returns kept by cell $i$ depends on *both* the network topology and $\beta$. When $\beta = 0$ there is no sharing, and when $\beta = 1$ cells share everything equally among all connections and themselves. We refer to $\beta$ as interaction strength. A given group topology (unweighted adjacency matrix) and $\beta$ completely specify $\mathbf{c}$.

Within a group of $N$ cells, the overall returns on viability that a given cell enjoys, then, comprises its own returns as well as whatever is shared with it by other members of the group. This can be written as $\tilde{v}_i = v_i^\alpha c_{ii} + \sum_{j \neq i}^{n} v_j^\alpha c_{ji}$, or equivalently, $\tilde{v}_i = \sum_j^n v_j^\alpha c_{ji}$. Note that this is a column sum, since it describes the total *incoming* viability returns a cell receives as a result of its own effort and trade with neighboring cells. Therefore, we write the group level reproduction rate (i.e. the group fitness) for a group of $N$ cells as

$$
\begin{aligned}
W &= \sum_{i=1}^{i=N} \tilde{b}_i \cdot \tilde{v}_i \\
W &= \sum_{i=1}^{i=N} \tilde{b}_i \sum_{j=1}^{j=N} v_j^\alpha c_{ji} \\
W &= \sum_{i=1}^{i=N} \sum_{j=1}^{j=N} b_i^\alpha c_{ji} v_j^\alpha,
\end{aligned}
\tag{1}
$$

where all three of the above equations are equivalent. We investigate evolutionary outcomes under this definition of group level fitness for groups with different topologies (who shares with whom), and in scenarios with various return on investment exponents $\alpha$.

## Results

### Fixed resource sharing

We first consider cases wherein cells within a group share across fixed intercellular interactions. In each case we vary the return on investment exponent, $\alpha$, between 0.5 and 1.5, and the interaction strength, $\beta$, between 0.0 and 1.0, both in increments of 0.1. For each combination of topology, $\alpha$, and $\beta$, the group investment strategy ($v_i$ for all $i$) was allowed to evolve for 1000 generations.

We begin with simple topologies: groups with no connections and groups that are maximally connected. They represent, respectively, the case in which all cells within the group are autonomous and the case in which every cell interacts with all others (i.e. a 'well-mixed' group). In the absence of interactions, cells cannot benefit from functions performed by others and therefore must perform both functions $v$ and $b$; hence specialization is not favored, and does not evolve. In the fully connected case, a high degree of specialization is observed for many values of $\alpha$ and $\beta$ (*Figure 1a*). Consistent with classic results (*Cooper and West, 2018*; *Michod et al., 2006*; *Ispolatov et al.,*

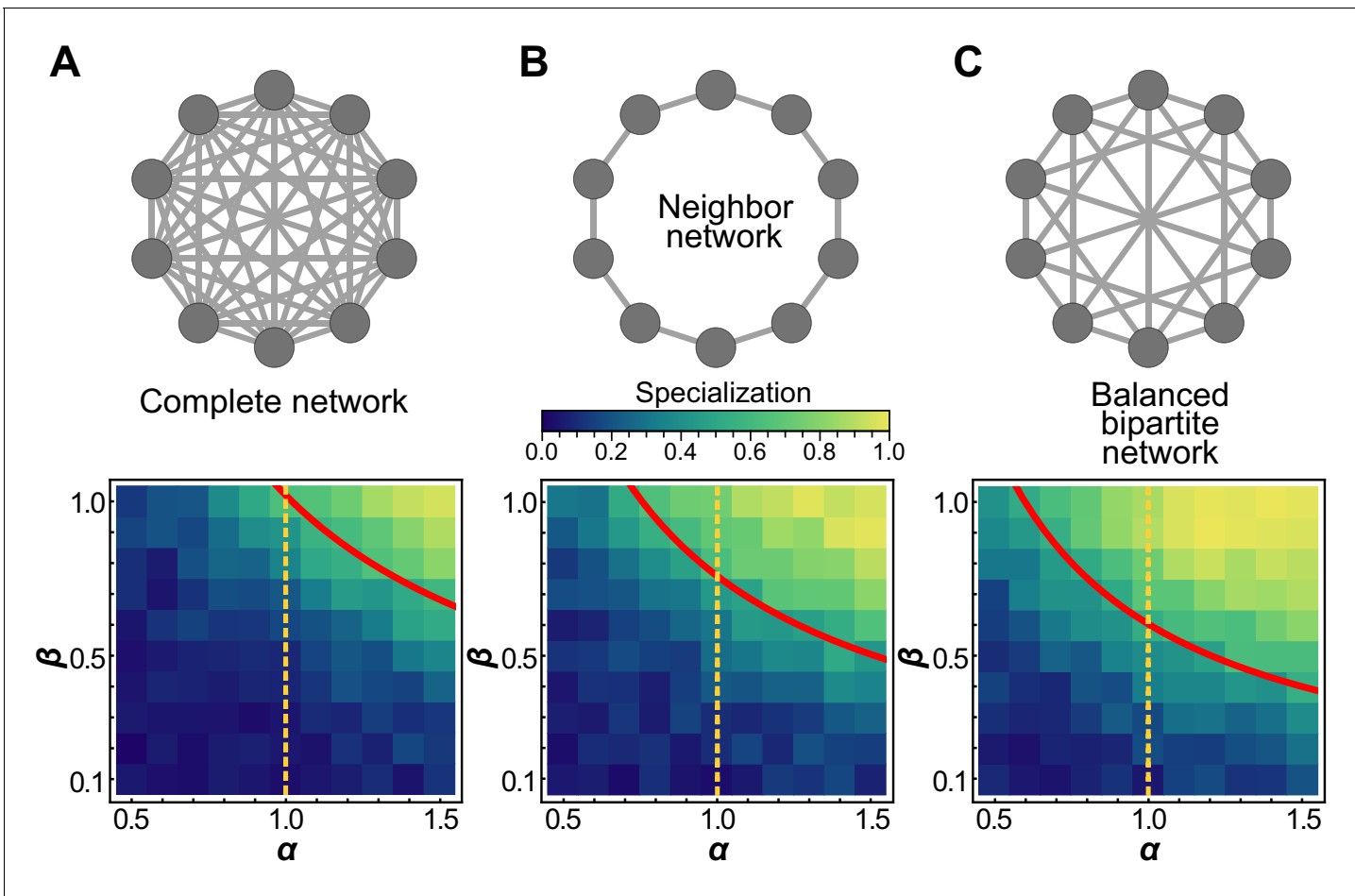

**Figure 1.** Schematic of topology for a simplified ten cell group (first row), and mean specialization as a function of specialization power $\alpha$ and interaction strength $\beta$ across the entire population. (**A**) When each cell in the group is connected to all others, specialization is favored only when $\alpha>1$. (**B**) For the nearest neighbor topology, specialization is favorable for a wider range of parameters, including for some values of $\alpha<1$. Specifically, specialization is advantageous when $\alpha>\frac{3}{4\beta}$. (**C**) Connecting alternating specialists creates a bipartite graph which maximizes the benefits of specialization and the range of parameters for which it is advantageous. In this case, specialization is favorable wherever $\alpha>\frac{3}{5\beta}$. The red curves represent analytical predictions for $\alpha^*$, the lowest value of $\alpha$ for which complete generalization is disfavored, and the orange vertical lines are at $\alpha=1$ to guide the eye. While analysis shows that *some* degree of specialization must occur in the regime upward and to the right of the red curves, simulations reveal that when complete generalization is disfavored complete specialization *is* favored in these networks.

*2012*; *Solari et al., 2013*; *Michod, 2007*; *West et al., 2015*), specialization is only achieved in the fully connected case for $\alpha>1$.

Next, we consider a simple sparse network in which each cell within a group is connected to only two other cells, forming a complete ring (*Figure 1b*); we refer to this as the neighbor network. Surprisingly, preventing trade between most cells encourages division of labor. We find that specialization evolves even when $\alpha<1.0$, that is, when the returns on investment are saturating or concave. In our simulations, this topology leads to alternating specialists in viability and fecundity (*Figure 1b*). Analytically, we find that this topology always favors at least some degree of specialization whenever $\alpha>\frac{3}{4\beta}$.

We next study a network with cells that can be separated into two disjoint sub-groups, where every edge of the network connects a cell in one sub-group to a cell in the other sub-group and no within sub-group connections exist, that is, a bipartite graph (*Figure 1c*). We refer to the specific network structure in *Figure 1c* as the 'balanced bipartite' network. We find that specialization evolves even when $\alpha<1.0$, similar to the neighbor network. However, we find that specialization evolves for a wider range of $\alpha$ and $\beta$ values for the balanced bipartite network than for the neighbor network.

We can analytically determine under what conditions complete generalization is optimal. The complete generalist investment strategy is where every cell in the group invests equally into viability and fecundity, defined as: $v_i^* = \frac{1}{2}$ for all $i$. For these simple topologies, the complete generalist strategy is either a maximum or a saddle point, depending on the values of $\alpha$ and $\beta$. Complete generalization is only favored when the Hessian evaluated at the generalist investment strategy $\frac{\partial^2 W}{\partial v_k \partial v_\ell}\big|_{\vec{v}^*} = \mathbf{H}^*$ is negative definite, that is, all of its eigenvalues are negative. The largest eigenvalues of the Hessian for the complete, neighbor, and balanced bipartite networks are $\alpha\left(\frac{1}{2}\right)^{2\alpha-3}(-1+\alpha\beta)$, $\alpha\left(\frac{1}{2}\right)^{2\alpha-3}(-1+\frac{4}{3}\alpha\beta)$, and $\alpha\left(\frac{1}{2}\right)^{2\alpha-3}(-1+\frac{2N}{N+2}\alpha\beta)$, respectively. When $\alpha$ and $\beta$ are chosen so that the largest eigenvalue becomes non-negative, complete generalization cannot maximize group fitness.

While we have not analytically shown where the fitness maximum occurs in cases where the generalist strategy becomes a saddle point, evolutionary simulations (*Figure 1*) suggest that when complete generalization is not a fitness maximum, a high degree of (or even complete) specialization typically *does* maximize fitness.

In all cases in which complete specialization is achieved in evolutionary simulations, $\tilde{v}\tilde{b}$ terms for viability specialists go to zero, as they cannot reproduce on their own. Furthermore, the fecundity specialists are entirely reliant on the viability specialists for their survival; if viability sharing were suddenly prevented, their $\tilde{v}\tilde{b}$ terms would also be zero. This amounts to complete reproductive specialization (*Cooper and West, 2018*; *Kirk, 2005*; *Michod, 2006*).

## Evolving resource sharing

Until now, sharing has been included in every intercellular interaction within groups. Here, we consider the case in which there is initially no sharing, and sharing must evolve along with specialization. These simulations begin with no resource sharing (i.e. $\beta = 0$); during every round, each group in the population has a 2% chance that a mutation will impact its developmental program, and the $\beta$ value for one of its cells will change. The new $\beta$ value is chosen from a truncated Gaussian with standard deviation of 10% of the mean, centered on the current value. Whatever is not retained is shared equally across all interactions, including the self term.

Evolutionary simulation results are similar to those from the fixed-sharing model (*Appendix 1— figure 1*). Saturating specialization (i.e. specialization despite a concave return function) still occurs for the neighbor and balanced bipartite topologies. Thus, for both fixed and evolved resource sharing, we observe specialization for the largest range of parameters (including $\alpha<1$) not when the group is maximally connected, but rather when connections are fairly sparse. Therefore, a sparse group topology constitutes a cooperation-prone physical substrate that can favor the evolution of cellular.

As an example of the benefit of evolving sharing, consider that the maximum fitness according to *Equation 1* for a group of $N$ disconnected cells scales as $N\left(\frac{1}{2}\right)^{2\alpha}$. On the other hand, for the balanced bipartite network with a complete specialization strategy (i.e. $\vec{v} = \langle 0, 1, 0, 1, ...\rangle$), the fitness scales as

$\left(\frac{N^2\beta}{2N+2}\right)$. The ratio of these fitnesses is $\left(\frac{N^2\beta}{2N^2+2N}\right)2^{2\alpha} \approx \beta 2^{2\alpha-1}$, where the approximation is for large $N$. So for larger groups and when $\alpha > \frac{1}{2} - \frac{\log\beta}{2\log 2}$, if a group can evolve resource sharing (i.e. letting $\beta \to 1$ and adopting the specialist investment strategy) its maximum fitness will increase.

## Benefit of specialization

We now consider a simple example to highlight why specialization can be adaptive despite saturating (i.e., concave) returns from trade. Consider groups of four cells, connected via the nearest-neighbor topology (i.e. in a ring). We directly calculate the group-level fitness of generalists and specialists for two scenarios: $\alpha = 0.9$ and $\alpha = 1$ by summing the contributions of each cell within these groups (*Figure 2*). In this simple scenario, reproductive specialization strongly increases group fitness (33% for $\alpha = 1$ and 16% for $\alpha = 0.9$).

The benefit of specialization in neighbor networks increases with group size. For a ring of size $N$, fitness under the specialist strategy $\vec{v} = \langle 0, 1, 0, 1 ... \rangle$ is $W = \frac{\beta}{3}N$. For a ring of generalists the fitness is $W = N(\frac{1}{2})^{2\alpha}$. Therefore, whenever $\alpha > \frac{\log 3 - \log \beta}{2\log 2}$, the ring of complete specialists enjoys a greater fitness than the ring of complete generalists. Again, note that complete generalization becomes disfavored when $\alpha > \frac{3}{4\beta}$, so there *is* a narrow regime where $\frac{3}{4\beta} < \alpha < \frac{\log 3 - \log\beta}{2\log 2}$ during which neither complete generalization nor complete specialization is optimal. Numerical optimization and evolutionary simulations

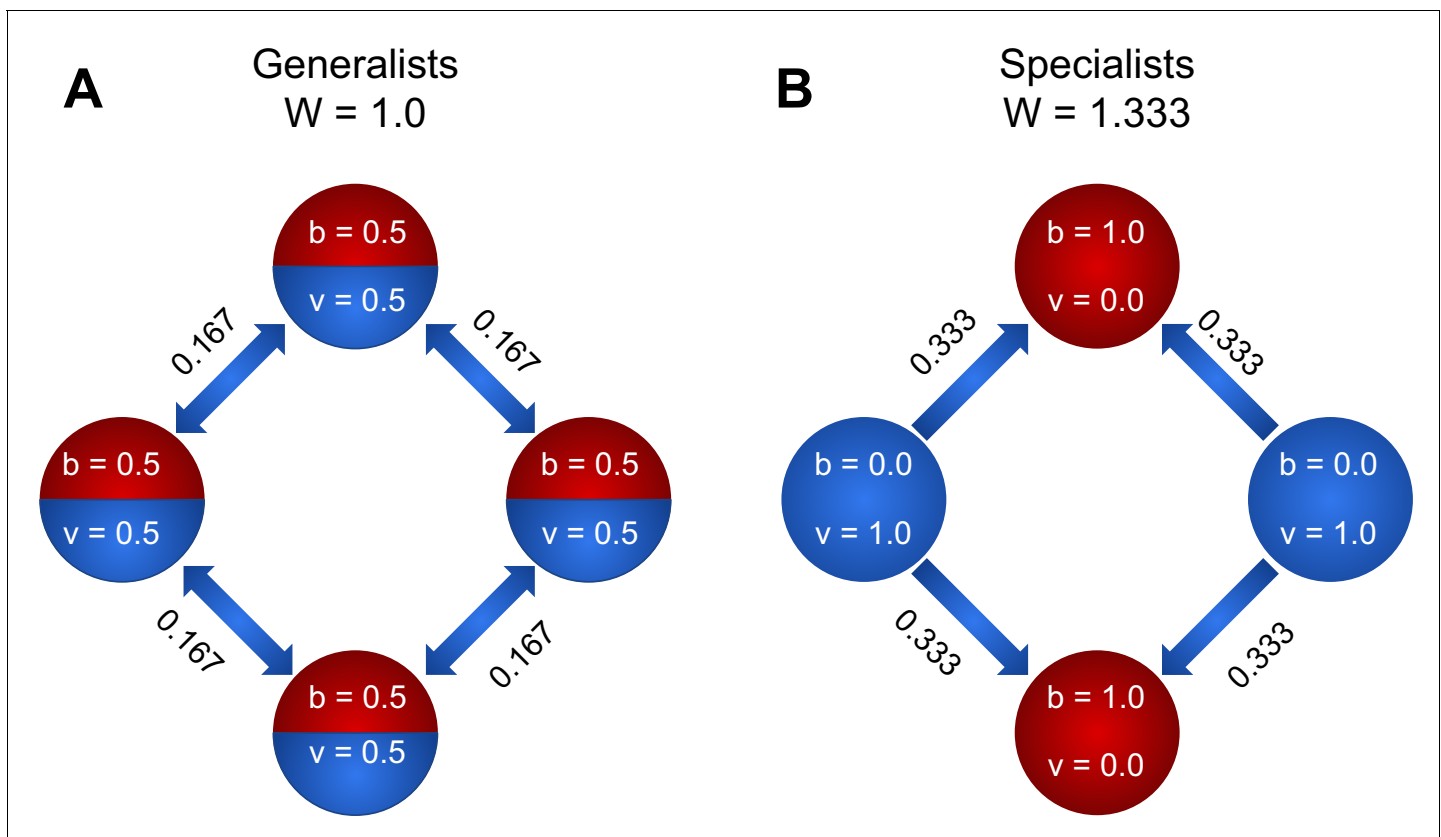

**Figure 2.** To explore how specialization can be favored by the nearest-neighbor topology, we compare the fitness of a four member system when cells are (A) generalists and (B) specialists. We first consider the case of linear functional returns ($\alpha = 1$). For the case of generalists (A), each cell receives as much viability as it shares, and all nodes contribute equally to the fitness of the group. Therefore, the fitness of the group is $W = 4 \cdot \frac{1}{2} \cdot \frac{1}{2} = 1$. For the case of specialists, however, the viability specialist cells (blue) have $\tilde{v}\tilde{b} = 0$, while the fecundity specialist cells have nonzero $\tilde{v}\tilde{b}$ due to the fact that they receive $\frac{1}{3}$ of each viability specialist's output. Thus the fitness of the group is $W = 2(2 \cdot \frac{1}{3}) = \frac{4}{3}$. Thus, fitness is higher for the group of specialists, so specialization is favored. For $\alpha = 0.9$, the fitness of generalists is 1.15, and the fitness of specialists is 1.33. Thus, even though the returns on investment are saturating (i.e. concave), specialization is favored.

suggest that even in this region, however, the specialization score of the optimal strategy is large (**Figure 1**).

## Effect of sparsity

Surprisingly, saturating specialization appears to be the rule, rather than the exception, for sparsely connected graphs. We investigated Erdős-Rényi random graphs with varying degrees of connectivity to systematically examine the relationship between sparsity and the value of $\alpha$ at which specialization is favored. We find that many randomly assembled graphs obtain maximum fitness through complete reproductive specialization even when $\alpha$ is below 1 (**Figure 3b,c**). It is only at the extremes of sparsity and connectivity (near the fully connected or fully unconnected points) that generalists maintain superior fitness for all values of $\alpha<1$. We further show that this general trend is independent of the size of a group; saturating specialization is favorable for groups of size $N = 10$, $N = 100$, and $N = 1000$. When network connectivity is at its minimum, the group consists solely of isolated cells that cannot interact. Under these conditions generalists are favored. Similarly, at maximum connectivity every cell interacts with every other cell. Under these conditions generalists are favored unless $\alpha\beta>1$. However, when connectivity is small but not zero, specialization arises most readily. We conjecture that the troughs in **Figure 3b**, where specialization occurs for the lowest values of $\alpha$, occur when connectivity is just large enough so that the existence of a spanning tree is more likely than not.

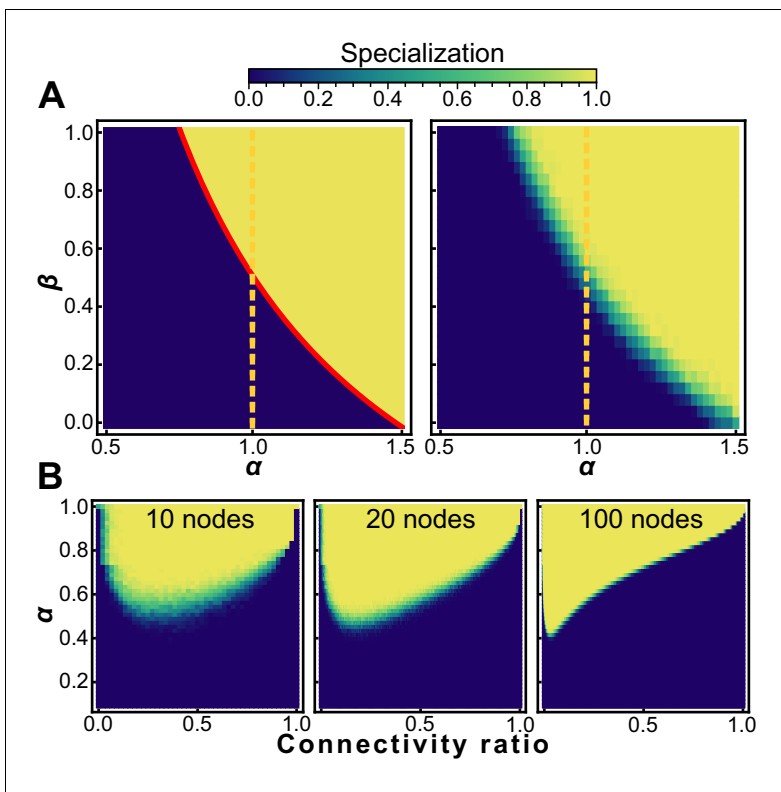

**Figure 3.** Sparsity encourages specialization. Heat maps showing conditions that favor specialists (white) and generalists (black) for nearest neighbor topologies (**A**, left) and randomly generated graphs with the same connectivity as nearest neighbor topologies (**A**, right). Specialization is adaptive on a neighbor network for $\alpha>\frac{3}{4\beta}$; random networks with the same mean connectivity as the nearest neighbor topology behave similarly. (**B**) The sparsity of a random graph affects how likely it is to favor specialization. We numerically maximize fitness for random graphs of size $N = 10$ (left), $N = 20$ (middle), and $N = 100$ (right) at different levels of sparsity, and subsequently measure the specialization $\mathcal{S}$ of the fitness maximizing investment strategy. The horizontal axis is the fraction of possible connections present ranging from 0 (none) to 1 (all). The vertical axis is the specialization power $\alpha$, and the colormap shows mean specialization.

## Filaments and trees

Sparse topologies like the neighbor network configuration have significant biological relevance, and direct ties to early multicellularity. The first step in the evolution of multicellularity is the formation of groups of cells (*Szathmáry and Smith, 1995*; *Kirk, 2005*; *Willensdorfer, 2008*; *Bonner, 1998*; *Fairclough et al., 2010*). Simple groups readily arise through incomplete cell division, forming either simple filaments (*Figure 4a*) or tree-like morphologies (*Figure 4b*; *Bengtson et al., 2017b*; *Droser and Gehling, 2008*; *Berman-Frank et al., 2007*; *Ratcliff et al., 2012*). Filament topologies have been widely observed in independently-evolved simple multicellular organisms, from ancient fossils of early red algae (*Butterfield, 2000*; *Figure 4a*) to extant multicellular bacteria (*Claessen et al., 2014*) and algae (*Umen, 2014*). Branching multicellular phenotypes have also been observed to readily evolve from baker's yeast (*Ratcliff et al., 2015*; *Figure 4b*), and are reminiscent of ancient fungus-like structures (*Bengtson et al., 2017a*) and early multicellular fossils of unknown phylogenetic position from the early Ediacaran (*Droser and Gehling, 2008*).

Simulations of populations of groups with filamentous and branched topologies reveal that specialization is indeed favored in the sub-linear regime (*Figure 4a and b*); conversely, sub-linear specialization is never observed for fully connected topologies (*Figure 4c*). While the generalist strategy is never a critical point for these networks (which have $\mathbf{c} \neq \mathbf{c}^T$, see Materials and methods), we conjecture that there is a nearby critical point which maximizes fitness at small values of $\alpha$ and becomes unstable at larger values of $\alpha$. We introduce a new metric, $\alpha^*$, defined as the value of $\alpha$ such that the largest (least negative) eigenvalue of the Hessian evaluated at the complete generalist strategy is zero when $\beta = 1$. For topologies in which each member has the same number of neighbors, $\alpha^*$ is a critical value at which generalization is no longer an optimal strategy. However, even for groups

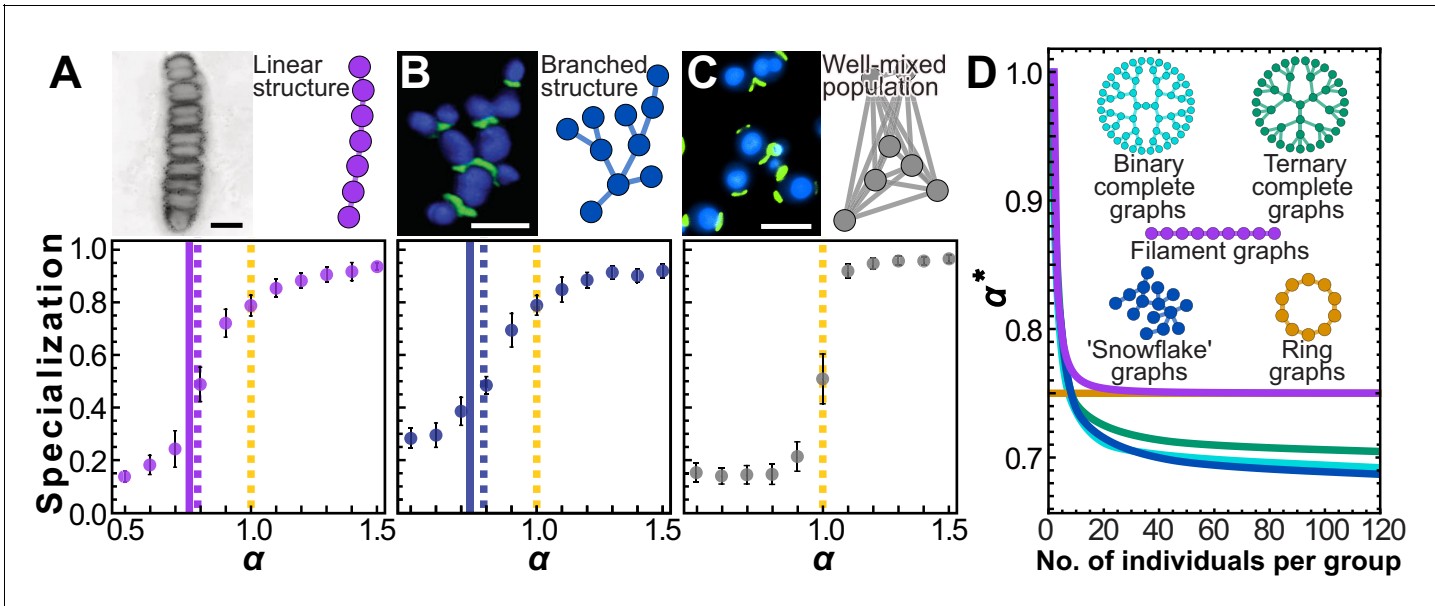

**Figure 4.** Simple multicellular organisms with sparse topologies. We show two examples of simple multicellular organisms with linear and branched topologies. The image in (**A**) is a fossilized rhodophyte specimen of *Bangiomorpha pubescens*, courtesy of Prof. Nicholas Butterfield (see e.g. *Butterfield, 2000*); the image in (**B**) is a confocal image of 'snowflake yeast' showing cell volumes in blue and cell-cell connections in green; the image in (**C**) is an epifluorescence image of individual yeast cells from a planktonic culture, with the same staining technique as in (**B**). Scale bars in pictures = 10 μm. Panels include cartoons depicting simplified topologies. Topologically similar to the two-neighbor configuration, these configurations yield similar simulation results. Specialization is plotted as a function of $\alpha$. Solid (A) and blue (B) vertical lines (A and B) indicate analytical solutions for the transition point where the Hessian evaluated at $\vec{v} = \frac{1}{2}\vec{1}$ stops being negative definite, that is, $\alpha^*$; dotted lines indicate roughly where the simulation curves cross specialization of 0.5, that is, the 'true' transition value of $\alpha$ where specialization becomes favored. (**C**) In contrast, for a well-mixed group with fully connected topology, $\alpha^* = 0.5$, indicating specialization only occurs when there are accelerating returns on investment. (**D**) To further explore trees and filaments we analytically solved for $\alpha^*$ for various types of trees and filaments of different sizes. $\alpha^*$ is plotted versus group size for several topologies. This is a proxy measure of how amenable a network structure is to specialization.

Prof. Butterfield has granted permission to distribute the image in panel A under the terms of a Creative Commons Attribution license [https://creativecommons.org/licenses/by/4.0/]; further reproduction of this image should adhere to the terms of the CC BY 4.0 license with an attribution to Prof. Butterfield.

where the number of neighbors for each cell varies, we can still use $\alpha^*$ as a proxy for how amenable a topology is to saturating specialization. The smaller $\alpha^*$, the more specialization is likely to be favored. We plot vertical lines where $\alpha = \alpha^*$ (solid lines in *Figure 4(a) Figure 4(b)*), and dotted lines to indicate roughly where the simulation curves cross specialization of 0.5. These results show that, for these topologies, $\alpha^*$ acts as an effective metric for how amenable a network is to saturating specialization. This metric $\alpha^*$ only depends on topology and can in principle be calculated analytically given any network. We examined the value of $\alpha^*$ as filaments and a variety of tree-like structures grow larger, and find that specialization becomes more strongly favored (*Figure 4D* ). While group size has no effect on specialization for some topologies, like the neighbor network, filaments and trees all see a decrease in $\alpha^*$ as group size increases; $\alpha^*$ eventually plateaus once groups are larger than a few tens of cells. Simple and easily accessible routes to multicellular group formation can readily evolve in response to selection for organismal size (*Ratcliff et al., 2012*), and this process may also strongly favor the evolution of cellular differentiation (*McCarthy and Enquist, 2005*; *Heim et al., 2017*; *McClain and Boyer, 2009*; *Bonner, 1998*).

## Mean field model

Finally, to capture some general principles underlying this phenomenon, we consider a mean-field model with $N$ cells ($N >> 1$), each of which is connected to $z$ other cells. For simplicity we consider the case in which $\beta = 1$ and $\alpha = 1$. We pick $\alpha = 1$ as at this point, if the fitness of specialists is greater than that of generalists, specialization will be favored for at least some values of $\alpha < 1$. If the fitness of generalists is greater than or equal to that of specialists, specialization will only be favored if $\alpha > 1$.

For generalists, the fitness is simply $W_G = N/4$, as each cell has $v = 1/2$ and $b = 1/2$ (before and after sharing). Viability specialists produce $v = 1$ and $b = 0$, while fecundity specialists produce $v = 0$ and $b = 1$. Viability specialists then share $v = 1/(z+1)$ with each of their $z$ neighbors. After sharing, fecundity specialists receive $v = 1/(z+1)$ from each of their viability specialist neighbors. But how many of their neighbors are viability specialists? We label the fraction of cells connected to fecundity specialists that are viability specialists $f$, that is, $f$ is the mean number of viability specialists connected to each fecundity specialist divided by $z$, averaged over all fecundity specialists. For a bipartite graph, $f = 1$; for a randomly connected graph on which half of cells are viability specialists and half of cells are fecundity specialists, $f = 1/2$. Group fitness is thus:

$$W_S = \frac{zfN}{2(z+1)}. \tag{2}$$

Here, $zf/(z+1)$ is the average viability returns each fecundity specialist has received after sharing, which is multiplied by the amount of fecundity each fecundity specialist has (1) and the number of fecundity specialists ($N/2$). Writing $W_S$ in terms of $W_G$:

$$W_S = \frac{2zfW_G}{z+1}. \tag{3}$$

Specialists will be favored if the ratio $W_S/W_G > 1$. This will be true if:

$$f > \frac{z+1}{2z}, \tag{4}$$

which reduces to:

$$f > \frac{1}{2} + \frac{1}{2z}. \tag{5}$$

This inequality implies that specialization will only be favored if fecundity specialists are preferentially connected to viability specialists, that is, if $f > 1/2$. Further, for a fully connected network $f = 1/2$, so this inequality is never satisfied, that is, specialists cannot have larger fitness than generalists for $\alpha = 1$ and fully connected topologies, as classically predicted.

Further, $f$ cannot be more than 1, so if the threshold from the inequality in *Equation 5* is greater than or equal to 1, specialization cannot be favored for $\alpha < 1$. Thus, specialization for $\alpha < 1$ is only possible if:

$$\frac{1}{2}+\frac{1}{2z}<1, \tag{6}$$

which reduces to: $z>1$. This again reproduces a classic result: specialization for $\alpha<1$ is not possible for disconnected cells.

This analysis allows us to interrogate specific cases. For example, if $z=3$, $f$ must be greater than 2/3, while if $z=4$, $f$ must only be greater than 5/8. Can such networks be constructed? The answer will depend on both the number of cells and how they are connected. Ultimately, the question of if a graph can be made with particular values of $f$ and $z$ is a graph coloring problem, and beyond the scope of this manuscript. However, this inequality presents a useful heuristic which can be used to determine if specialization is favored by measuring just a few properties of the graph.

## Effect of varying ratios of specialists

We now allow the fraction of fecundity specialists to be $X$ (rather than forcing $X=1/2$). For generalists, the group fitness is unchanged, $W_G = N/4$, while for specialists the group fitness is:

$$W_S = \frac{zfXN}{z+1}. \tag{7}$$

Writing $W_S$ in terms of $W_G$ gives:

$$W_S = \frac{4zfXW_G}{z+1}. \tag{8}$$

Specialists will be favored if the ratio $W_S/W_G>1$. This will be true if:

$$f>\frac{z+1}{4Xz}=\frac{1}{4X}+\frac{1}{4Xz}. \tag{9}$$

Compared to the threshold value of $f$ when $X=1/2$, if $X>1/2$, that is, more than half of cells are fecundity specialists, the value of $f$ necessary for specialization to be favored is lower. If $X<1/2$, the threshold value of $f$ is higher than if $X=1/2$. In other words, 1:2 is different from 2:1, and they both are different from 1:1. Once again, the question of if a particular configuration can be created–and how–is a graph coloring problem beyond the scope of this manuscript. However, this mean field heuristic gives us some information about how to expect graphs with different ratios of specialists to generalists to behave.

We again ask what must be true for $f$ to be less than 1 (if $f>1$, specialization will not be favored). Thus, specialization is only possible if:

$$\frac{1}{4X}+\frac{1}{4Xz}<1, \tag{10}$$

which reduces to:

$$X>\frac{1}{4}+\frac{1}{4z}. \tag{11}$$

For a mean field model, specialization with $\alpha<1$ is impossible if fewer than one fourth of cells are fecundity specialists. We stress here that this is a mean field model, and does not apply to scenarios in which cells have a wide range of values of $z$. If such networks do or do not favor specialization for $\alpha<1$ will again be a graph coloring problem.

## Discussion

During the evolution of multicellularity, formerly autonomous unicellular organisms evolve into functionally-integrated parts of a new higher level organism (*West et al., 2015*; *Michod and Nedelcu, 2003*). Evolutionary game theory (*Corning and Szathmáry, 2015*; *Nash, 1950*; *Smith, 1988*) argues that functional specialization should only evolve when increased investment in trade increases reproductive output. Conventionally, this requires returns from specialization to be accelerating, that is, convex or super-linear (*Szathmáry and Smith, 1995*; *Smith and Szathmáry, 1997*; *Goldsby et al.,*

*2012*; *Corning and Szathmáry, 2015*; *Boza et al., 2014*; *Taborsky et al., 2016*; *Page et al., 2006*; *Rueffler et al., 2012*; *Szekely et al., 2013*). While this idea is intuitive, it is, in the case of fixed group topology, also overly restrictive. In this paper, we explore how social interactions within groups, measured by their network topology, affect the evolution of reproductive specialization. Indeed, when all cells within groups interact (with equal interaction strength), returns on investment must be an accelerating, that is, convex, function of investment for specialization to evolve (*Figure 1a*; *Szathmáry and Smith, 1995*; *Smith and Szathmáry, 1997*; *Corning and Szathmáry, 2015*; *Cooper and West, 2018*). Yet for a broad class of sparsely connected networks, complete specialization can evolve even when the viability and fecundity return on investment curves are saturating, that is, concave (*Figure 3*).

To understand how specialization can be favored despite concave return on investment (ROI) curves, consider Jensen's inequality. Jensen's inequality states that for a convex function $F(x)$, $\langle F(x) \rangle > F(\langle x \rangle)$, that is, the average value of $F(x)$, $\langle F(x) \rangle$, is larger than $F(\langle x \rangle)$, where $\langle x \rangle$ is the average value of $x$. A corollary of Jensen's inequality is that the opposite is true for concave functions, that is, for a concave function $G(x)$, $\langle G(x) \rangle < G(\langle x \rangle)$. Jensen's inequality guarantees that for concave ROI functions generalists produce more total viability and fecundity than specialists, and that for convex ROI functions specialists produce more total viability and fecundity than generalists.

Crucially, however, Jensen's inequality does not connect ROI convexity/concavity to group fitness. Jensen's inequality relates the degree of specialization to the average viability and average fecundity produced, but does not itself say anything about group fitness, which is the product of viability and fecundity averaged across all cells. For fully connected topologies (i.e. *Figure 4c*), greater absolute productivity proportionally increases group fitness, and differentiation can only evolve with accelerating benefits of specialization. This is not the case for topologically structured organisms, where fitness also depends on how complementary specialist cells are connected. Natural selection acts on realized productivity, that is, average $vb$; mutations that increase average $v$ or average $b$ without increasing average $vb$ are not adaptive. The importance of connecting complementary specialists has long been appreciated in other contexts, such as metabolic cross-feeding, for which it has been shown that the spatial arrangement of unlike specialists plays a key role in determining their productivity (and thus fitness) (*Co et al., 2020*). Indeed, While Jensen's inequality ensures that generalists will produce more viability and fecundity than specialists given a concave ROI function, specialization can still increase the fitness of topologically structured groups by increasing realized productivity.

Rather than being unusual, networks favoring specialization readily arise as a consequence of physical processes structuring simple cellular groups (*Allen et al., 2017*). For example, septin defects during cell division create multicellular groups with simple graph structures (*Figure 4a and b*), where cells are connected only to parents and offspring (*Bengtson et al., 2017b*; *Droser and Gehling, 2008*; *Ratcliff et al., 2012*; *Ratcliff et al., 2013*). If cells share resources only with physically-attached neighbors, then the physical topology of the group describes its interaction topology, and these sparse networks strongly favor reproductive specialization. Finally, we note that the primary benefit of sparsity is that sparse networks are likely to be at least somewhat bipartite. The more bipartite-like a network is, the less effort is wasted, and the easier it is for specialization to be favored.

Disentangling the evolutionary underpinnings of ancient events is notoriously difficult. Still, it is worth examining the independent origins of complex multicellularity, which are independent runs of parallel natural experiments in extreme sociality. Complex multicellularity (large multicellular organisms with considerable cellular differentiation) has evolved in at least five eukaryotic lineages, once each in the animals (*King, 2004*), land plants (*Kenrick and Crane, 1997*), and brown algae (*Silberfeld et al., 2010*), two or three times in the red algae (*Cock and Collén, 2015*; *Yoon et al., 2006*), and 8–11 times in fungi (*Nagy et al., 2018*). In all cases other than animals, these organisms form multicellular bodies via permanent cell-cell bonds, creating long-lasting highly structured cellular networks. Both fossil and phylogenetic evidence suggests that early multicellular organisms in these lineages were considerably less complex, growing as relatively simple graph structures. For example, 1.2 billion year old red algae formed linear filaments of cells (*Butterfield, 2000*), basal multicellular charophyte algae formed circular sheets of cells radiating from a common center (*Kenrick and Crane, 1997*), the ancestor of the brown algae likely formed a branched haplostichous thallus that was either filamentous or pseudoparenchymatous (*Silberfeld et al., 2010*), and hyphal

fungi are primarily composed of linear chains of cells. Much less is known about the topology of animals prior to the evolution of cellular specialization. One hypothesis is that early metazoans resembled extant colonial choanoflagellates (*Fairclough et al., 2013*), the closest-living protistan relatives of the animals (*Fairclough et al., 2010*). Extant colony-forming choanoflagellates have evolved a variety of multicellular structures with sparse cellular topologies and permanent cell-cell bonds. For example, many species form branched, tree-like structures (*Leadbeater, 2015*), *Choanoeca flexa* grows as a sheet of cells (*Brunet et al., 2019*), and *Salpingoeca rosetta* can form either linear chains or rosettes in which the cells are connected via cytoplasmic bridges formed through incomplete cytokinesis (*Dayel et al., 2011*). While these growth forms are quite diverse, they all share characteristics (i.e. permanent cellular bonds and sparse topologies) that promote the evolution of cellular differentiation.

The main differences between our work and previous investigations of the effect of group topology on specialization is that we consider the productivity of groups as a whole, not the cells within them, and we consider situations of highly asymmetric sharing. Our approach is general, and can be applied to other systems of trade and specialization, so long as (1) only the aggregate productivity of the group (and not the particles within it) is maximized, (2) the productivity of each particle within the group is a multiplicative function of returns on investment into two (or more) tasks, and (3) there is an asymmetry in how products of those investments are shared. While in this work we have focused on reproductive division of labor, a process in which fecundity returns are not shared at all, we show in the supplement that as long as sharing of two goods is sufficiently asymmetric, specialization with saturating returns on investment can still be adaptive (*Appendix 1—figure 2*).

Finally, we note that alternative paths to specialization likely exist. For example, cells at different positions in a group may experience different local environments, which may produce cells with varied fecundity-viability trade-offs. A previous paper demonstrated that the evolution of specialization is favored if these 'positional effects' result in an initially heterogeneous population of cell types (*Tverskoi et al., 2018*). However, these positional effects were considered for the case of well-mixed groups (i.e. completely connected network topologies). We thus anticipate that future work examining the relationship between cellular interaction topology and cellular heterogeneity (as well as a wide range of complex and varied relationships between viability, fecundity, and multicellular fitness) will provide unique insight into the origin and diversity of multicellular forms.

## Conclusion

We explored the evolution of reproductive specialization in multicellular groups with various cellular interaction topologies. Our results demonstrate that group topological structure can play a key role in the evolution of reproductive division of labor. Indeed, within a broad class of sparsely connected networks, specialization is favored even when the returns from cooperation are saturating (i.e. concave); this result is in direct contrast to the prevailing view that accelerating (i.e. convex), returns are required for natural selection to favor increased specialization (*Cooper and West, 2018*; *Michod et al., 2006*; *Ispolatov et al., 2012*; *Solari et al., 2013*; *Michod, 2007*; *West et al., 2015*). Our results underscore the central importance of life history trade-offs in the origin of reproductive specialization (*Michod et al., 2006*; *Michod, 2007*; *Hammerschmidt et al., 2014*; *van Gestel and Tarnita, 2017*; *Noh et al., 2018*), and support the emerging consensus that evolutionary transitions in individuality are not necessarily highly constrained (*Ratcliff et al., 2012*; *Ratcliff et al., 2017*; *Fairclough et al., 2010*; *Brunet and King, 2017*; *Pennisi, 2018*; *Black et al., 2019*; *Rose et al., 2020*; *van Gestel and Tarnita, 2017*; *Black et al., 2019*; *Staps et al., 2019*; *Grosberg and Strathmann, 2007*).

## Materials and methods

### Analysis

The gradient of the fitness with respect to the group investment strategy $\vec{v}$, is

$$\frac{\partial W}{\partial \vec{v}} = \sum_{k=1}^{N} \hat{e}_k \alpha \left( v_k^{\alpha-1} \sum_{j=1}^{N} c_{kj}(1-v_j)^{\alpha} - (1-v_k)^{\alpha-1} \sum_{j=1}^{N} c_{jk} v_j^{\alpha} \right) \tag{12}$$

where $\hat{e}_k$ is a unit vector in the $k^{\text{th}}$ direction. First notice that if $\mathbf{c} = \mathbf{c}^T$, and $\vec{v} = \frac{1}{2}\vec{1}$ where $\vec{1}$ is a vector of ones, then the gradient is zero. This strategy, $\vec{v} = \frac{1}{2}\vec{1}$, corresponds to the 'generalist' strategy, where every cell invests equally into both tasks. Second, notice that if $\mathbf{c} \neq \mathbf{c}^T$ then the gradient is *not* zero under the generalist strategy, so at least some degree of specialization must be necessary to maximize fitness. To determine the stability of this solution we examine $\mathbf{H}^*$, the Hessian (see SI *Equation 3*) evaluated at the generalist critical point. If $\mathbf{H}^*$ is negative definite, then the generalist strategy is a fitness maximum and is therefore an optimal strategy. If, on the other hand, $\mathbf{H}^*$ has both positive and negative eigenvalues then the generalist strategy lies at a saddle point within the fitness landscape, and therefore the optimal strategy must be somewhere else in (or on the boundary of) the domain (i.e. $v_i \in [0, 1]$ for all $i \in 1, 2, ...N$). Finally, note that $\mathbf{H}^*$ is never positive definite since $\vec{1}$ is always an eigenvector with negative eigenvalue (when $\mathbf{c} = \mathbf{c}^T$).

We also use the zero crossing of the largest eigenvalue of $\mathbf{H}^*$ evaluated at $\vec{v} = \frac{1}{2}\vec{1}$ and $\beta = 1$ as an overall measure of how amenable a network is to specialization, even when $\mathbf{c} \neq \mathbf{c}^T$.

## Evolutionary simulations

Our evolutionary simulations maintain the same overall structure as the Wright-Fisher model: a discrete-time Markov chain framework with fitness-weighted multinomial sampling between generations and constant population size. Therefore we refer to them as Wright-Fisher evolutionary simulations. We initialize a population of $\mathcal{N} = 1000$ groups, each of group size $N = 10$, with uniform random investment strategies. We then let them evolve for 1000 generations, selecting offspring according to the relative fitness of each group (see *Equation 1*). At each generation, there is a 2% chance for a mutation to a given group's investment strategy $\vec{v}$. If a mutation occurs, a new investment strategy is selected from a truncated multivariate gaussian distribution centered at the current (pre-mutation) investment strategy and with standard deviation equal to $\frac{1}{10}\vec{v}$. After mutations each group's fitness is calculated according to *Equation 1*, and the population is ranked according to fitness. Finally, $\mathcal{N}$ groups are selected (with replacement) to populate the next generation, according to a multinomial distribution weighted by the groups' fitness ranks.

## Measuring specialization

To quantify the degree of specialization associated with a given group's optimal investment strategy—the one which maximizes the fitness—we introduce the following metric, which we refer to simply as 'Specialization':

$$\mathcal{S} = \frac{2}{N}\sum_{i=1}^{N}\left(\max(v_i, 1 - v_i) - \frac{1}{2}\right). \tag{13}$$

Specialization ranges from 0 (for groups consisting of cells investing equally in functions *v* and *b*) to 1 for groups consisting of cells investing exclusively in either function.

## Code availability

All evolutionary simulations and other computations associated with this work are available at github.com/dyanni3/topologicalConstraintsSpecialization (*Yanni, 2020*; copy archived at https://github.com/elifesciences-publications/topologicalConstraintsSpecialization).

# Additional information

## Funding

| Funder | Grant reference number | Author |
| --- | --- | --- |
| National Science Foundation | IOS-1656549 | William C Ratcliff<br>Peter J Yunker |
| National Institutes of Health | GM138030 | William C Ratcliff |
| National Science Foundation | BMAT-2003721 | Peter J Yunker |

The funders had no role in study design, data collection and interpretation, or the decision to submit the work for publication.

### Author contributions

David Yanni, Data curation, Software, Formal analysis, Validation, Investigation, Methodology, Writing - review and editing; Shane Jacobeen, Conceptualization, Data curation, Software, Formal analysis, Validation, Investigation, Methodology, Writing - original draft; Pedro Márquez-Zacarías, Visualization, Methodology, Writing - review and editing; Joshua S Weitz, Supervision, Methodology, Writing - review and editing; William C Ratcliff, Conceptualization, Resources, Supervision, Writing - original draft, Project administration, Writing - review and editing; Peter J Yunker, Conceptualization, Resources, Supervision, Funding acquisition, Methodology, Writing - original draft, Project administration, Writing - review and editing

### Author ORCIDs

Peter J Yunker https://orcid.org/0000-0001-8471-4171

### Decision letter and Author response

Decision letter https://doi.org/10.7554/eLife.54348.sa1
Author response https://doi.org/10.7554/eLife.54348.sa2

## Additional files

### Supplementary files

• Transparent reporting form

### Data availability

All evolutionary simulations and other computations associated with this work are available at https://github.com/dyanni3/topologicalConstraintsSpecialization (copy archived at https://github.com/elifesciences-publications/topologicalConstraintsSpecialization); all parameters used in the current study are specified so all simulations can be repeated exactly.

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

## Appendix 1

### Analysis

As described in the main text, the fitness for a group of $N$ individuals is defined as

$$W = \sum_{i=1}^{N} \sum_{j=1}^{N} b_i^{\alpha} c_{ji} v_j^{\alpha} \tag{1}$$

and the gradient of the fitness with respect to the group investment strategy $\vec{v}$, is

$$\frac{\partial W}{\partial \vec{v}} = \sum_{k=1}^{N} \hat{e}_k \alpha \left( v_k^{\alpha-1} \sum_{j=1}^{N} c_{kj}(1-v_j)^{\alpha} - (1-v_k)^{\alpha-1} \sum_{j=1}^{N} c_{jk} v_j^{\alpha} \right) \tag{2}$$

where $\hat{e}_k$ is a unit vector in the $k^{\text{th}}$ direction.

### Hessian

The Hessian $\frac{\partial^2 W}{\partial v_k \partial v_\ell}$ is

$$H_{kl} = \begin{cases} -\alpha^2 \left( v_k^{\alpha-1} c_{k\ell}(1-v_\ell)^{\alpha-1} + (1-v_k)^{\alpha-1} c_{\ell k} v_l^{\alpha-1} \right) & k \neq \ell \\[2ex] \begin{aligned} & -2\alpha^2 \left( v_k^{\alpha-1} c_{kk}(1-v_k)^{\alpha-1} \right) \\ & + \left( (\alpha)(\alpha-1) v_k^{\alpha-2} \sum_j c_{kj}(1-v_j)^{\alpha} \right) \\ & + \left( (\alpha)(\alpha-1)(1-v_k)^{\alpha-2} \sum_j c_{jk} v_j^{\alpha} \right) \end{aligned} & k = \ell \end{cases} \tag{3}$$

Of particular interest for us is the value of the Hessian at the generalist strategy when $\mathbf{c} = \mathbf{c}^T$. In that case

$$\mathbf{H}^* = \alpha \left( \frac{1}{2} \right)^{2\alpha-3} [-\alpha\beta\mathbf{a} + (\alpha\beta-1)\mathbf{I}]. \tag{4}$$

where $\mathbf{a}$ is the row-normalized adjacency matrix of the network. If $\mathbf{A}$ is the network's adjacency matrix then

$$a_{ij} = \frac{A_{ij}}{\sum_{j=1}^{N} A_{ij}}.$$

**The case when $\mathbf{c} = \mathbf{c}^T$**

As noted above, when $\mathbf{c} = \mathbf{c}^T$, the generalist strategy is always a critical point where $\frac{\partial W}{\partial \vec{v}} = 0$. To determine the stability of this solution we examine $\mathbf{H}^*$ (*Equation 4*). If $\mathbf{H}^*$ is negative definite, then the generalist strategy is a fitness maximum and is therefore an optimal strategy. If, on the other hand, $\mathbf{H}^*$ has both positive and negative eigenvalues then the generalist strategy lies at a saddle point within the fitness landscape, and therefore the optimal strategy must be somewhere else in (or on the boundary of) the domain (i.e. $v_i \in [0,1]$ for all $i \in 1, 2, ...N$). Finally, note that $\mathbf{H}^*$ is never positive definite (when $\mathbf{c} = \mathbf{c}^T$). Consider $\mathbf{H}^* \vec{1}$:

$$\mathbf{H}^* \vec{1} = \alpha \left( \frac{1}{2} \right)^{2\alpha-3} [-\alpha\beta\mathbf{a}\vec{1} + (\alpha\beta-1)\mathbf{I}\vec{1}]$$

$$\mathbf{H}^* \vec{1} = \alpha \left( \frac{1}{2} \right)^{2\alpha-3} \left[ -\alpha\beta\vec{1} + (\alpha\beta-1)\vec{1} \right]$$

$$\mathbf{H}^* \vec{1} = -\alpha \left( \frac{1}{2} \right)^{2\alpha-3} \vec{1}.$$

Note $\mathbf{a}\vec{1} = \vec{1}$ since $\mathbf{a}$ is row-normalized. Furthermore, $\alpha > 0$, so $\vec{1}$ is always an eigenvector of $\mathbf{H}^*$ with a negative eigenvalue.

We can next ask, under what conditions is $\mathbf{H}^*$ negative definite? This will depend on the group topology, the nonlinear returns on investment $\alpha$, and the interaction strength $\beta$. We examine three cases: the neighbor graph, the balanced bipartite graph, and the complete graph.

**Appendix 1—table 1.** Largest eigenvalue of the Hessian evaluated at the generalist critical point as a function of $\alpha$, $\beta$, and $N$ for three topologies.

When the group size $N = 4$, the balanced bipartite graph coincides with the neighbor graph, and indeed the eigenvalues agree. Similarly, when $N = 2$ the balanced bipartite graph coincides with the complete graph and the eigenvalues agree. The interesting domain of $\alpha\beta$ is $(0, 1]$, so for the complete graph $\mathbf{H}^*$ is always negative definite. However, the balanced bipartite and neighbor graphs show regions where the generalist strategy is *not* stable.

| Topology | Largest eigenvalue |
| --- | --- |
| neighbor graph | $\alpha\left(\frac{1}{2}\right)^{2\alpha-3}(-1 + \frac{4}{3}\alpha\beta)$ |
| balanced bipartite graph | $\alpha\left(\frac{1}{2}\right)^{2\alpha-3}(-1 + \frac{2N}{N+2}\alpha\beta)$ |
| complete graph | $\alpha\left(\frac{1}{2}\right)^{2\alpha-3}(-1 + \alpha\beta)$ |

When $\mathbf{c} = \mathbf{c}^T$, the matrix $\mathbf{H}^*$ is a special type of matrix called a circulant matrix, with well known properties. Its eigenvalues are given by the discrete Fourier transform of its first row. The $k^{\text{th}}$ eigenvalue is

$$\lambda_k = \sum_{j=0}^{N-1} H_{1j}^* e^{\frac{2\pi i}{N} jk}.$$

For the ring topology with $N = 10$, for example

$$\lambda_k = \alpha\left(\frac{1}{2}\right)^{2\alpha-3}\left((-1 + \frac{2\alpha\beta}{3}) - \frac{\alpha\beta}{3}e^{\frac{\pi i k}{5}} - \frac{\alpha\beta}{3}e^{\frac{9\pi i k}{5}}\right),$$

which has its maximum when $k = 5$,

$$\max_k \lambda_k = \alpha\left(\frac{1}{2}\right)^{2\alpha-3}(-1 + \frac{4\alpha\beta}{3}).$$

The maximum eigenvalue for the balanced bipartite and complete graphs can be computed similarly.

## Evolution of resource sharing

Here we model the co-evolution of sharing and specialization. We start with generalists that do not share at all. We then allow the amount of sharing and the degree of specialization to evolve. As described in the main text, during every round, each group in the population has a 2% chance that one if its cells will mutate and change how much 'viability' it shares. When this occurs, the fraction of its output to retain is chosen from a Gaussian with standard deviation of 10% centered on the current value. Whatever is not retained is shared equally across its interactions. The degree of specialization evolves as in simulations described in the main text.

Results are shown in *Appendix 1—figure 1*, for neighbor topologies, balanced bipartite topologies, and for a complete network.

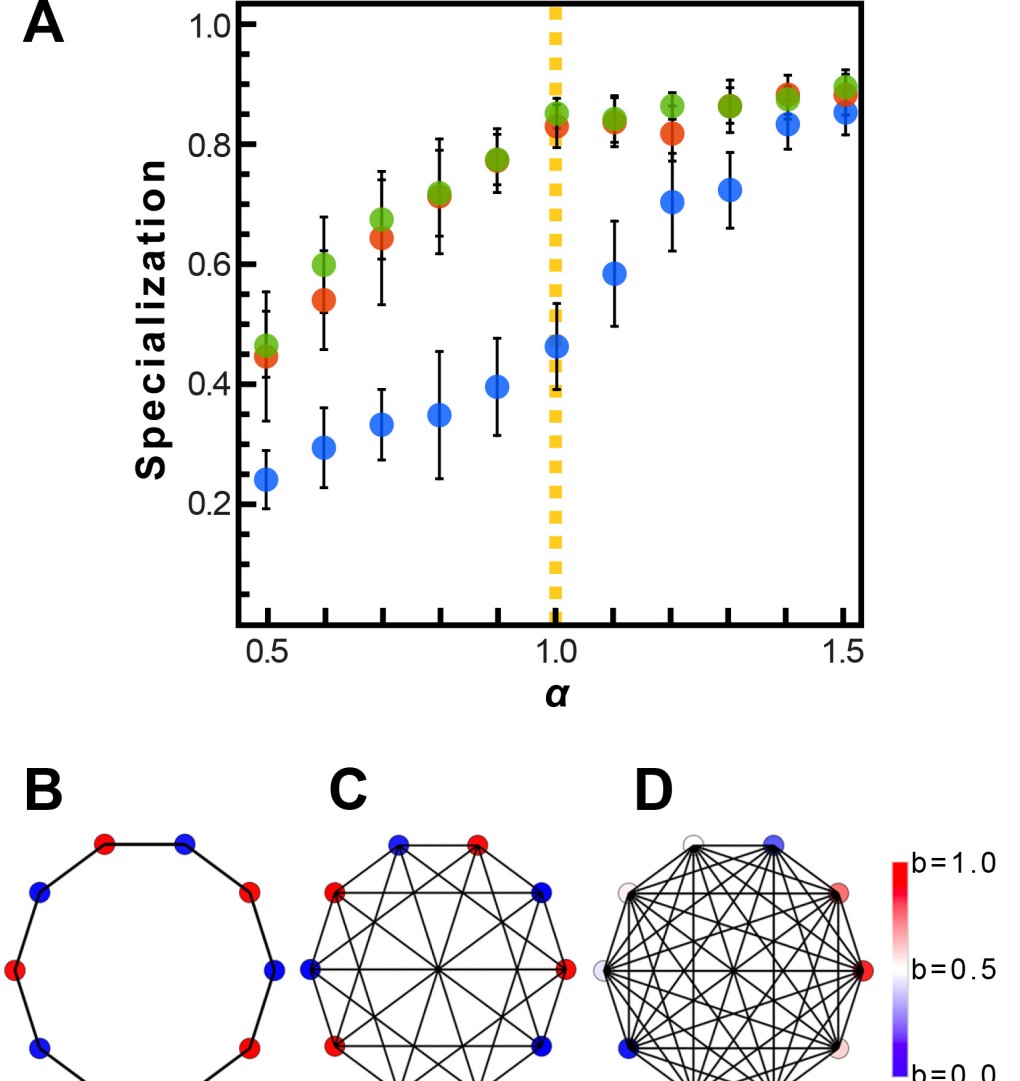

**Appendix 1—figure 1.** Evolution of resource sharing. (**A**) Initially, individuals do not share resources; however, they may evolve to do so via random mutations. Here, the mean specialization of the fittest of 100 groups each with 10 cells after 100,000 steps is plotted as a function of specialization power. Error bars are standard deviations across 10 replicates. Blue is the fully connected network, red is the neighbor network, and green is the balanced bipartite topology. (**B-D**) The final distribution of specialization values for individual cells in fully connected (**B**), nearest-neighbor (**C**), and balanced bipartite topologies (**D**). The color of cells in B-D represents their degree of specialization, as indicated in the scale bar.

## General case of sharing two resources

We have so far focused on reproductive specialization, wherein the returns from one type of task (reproduction) are completely unshared while returns from another task (viability) are shared according to some functional interaction strength $\beta$. Here, we generalize somewhat to consider the returns from two arbitrary tasks which may each be shared to some extent, given by functional interaction strengths $(\beta_1, \beta_2)$. For notational continuity we will continue to refer to the investment in those tasks as $b$ and $v$, and for tractability we will continue to assume that $\alpha_1 = \alpha_2$ and that there is a single topology governing who can trade with whom within the group. Of course, further generalizations could be made — e.g. each task could experience different returns on investment, there could be an arbitrary number of tasks, the availability of trading partners could differ between tasks, etc. However, we hope to show by this relatively modest generalization that there is nothing unique to

*reproductive* tasks whose fruits are totally unshared that leads to specialization under regimes of sublinear return on investment.

The fitness function is modified so that

$$W = (\mathbf{c_1}^T \cdot \vec{b}^\alpha) \cdot (\mathbf{c_2}^T \cdot \vec{v}^\alpha) \tag{5}$$

which yields the following Hessian at the generalist critical point (for the neighbor, balanced bipartite, and complete networks)

$$\mathbf{H}^*_{ab} = \alpha \left(\frac{1}{2}\right)^{2\alpha-2} \left( -2\alpha(\mathbf{c_1} \cdot \mathbf{c_2}^{\mathrm{T}})_{ab} + (\alpha-1)\left[ (\mathbf{c_1} \cdot \mathbf{c_2}^{\mathrm{T}} \cdot \vec{1})_a + (\mathbf{c_1} \cdot \mathbf{c_2}^{\mathrm{T}} \cdot \vec{1})_b \right] \delta_{ab} \right),$$

where

$$\mathbf{c_1} = \beta_1 \mathbf{a} + (1-\beta_1)\mathbf{I} \tag{6}$$

$$\mathbf{c_2} = \beta_2 \mathbf{a} + (1-\beta_2)\mathbf{I} \tag{7}$$

and,

$$a_{ij} = \frac{A_{ij}}{\sum_{j=1}^{N} A_{ij}},$$

where, as above, $\mathbf{A}$ is the graph's adjacency matrix (including self loops).

We see that for a given topology the adjacency matrix is fixed, so that $\mathbf{c_1}$ and $\mathbf{c_2}$ differ only in their functional interaction strengths $\beta_1$ and $\beta_2$. Therefore the maximum fitness strategy, specified by the vector $\vec{v}^*$, for a given group will depend under our model on the following parameters:

$\mathbf{A} \longrightarrow$ Adjacency matrix, specifies topology
$\beta_1 \longrightarrow$ Functional interaction strength of resource 1
$\beta_2 \longrightarrow$ Functional interaction strength of resource 2
$\alpha \longrightarrow$ Specialization power, assumed to be equal for resource 1 and 2

We demonstrate the effect of these parameters on the optimal strategy by finding the minimum value of $\alpha$ for which specialization becomes favored, which we denote $\alpha^*$, for a given pair $(\beta_1, \beta_2)$ and given topology. The results are shown in *Appendix 1—figure 2*.

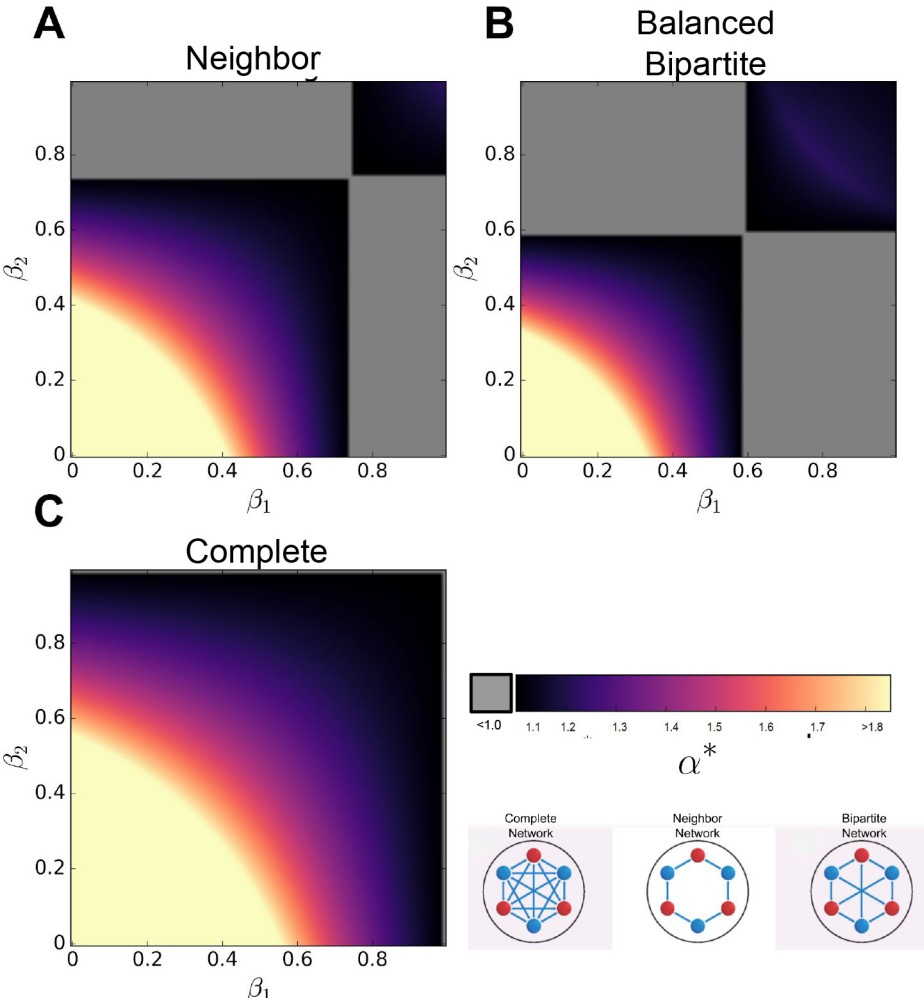

**Appendix 1—figure 2.** Effect of sharing two resources. When two resources are shared to different degrees, specified by $(\beta_1, \beta_2)$, specialization is sometimes favored under conditions of sublinear returns on investment $\alpha^* < 1.0$. Interestingly, specialization is favored when one resource is shared liberally while the other resource is shared sparingly (though it is not necessary to have one resource remain totally unshared).

## Jensen's inequality and sparse topologies and asymmetric sharing

To understand how average fitness decouples from average $v$ and average $b$ for sparse topologies and asymmetric sharing, consider a ring of four cells in three different configurations: one that alternates between viability and fecundity specialists, one in which like-specialists are connected to each other, and one in which all cells are generalists (pictured below). For simplicity, we will set $\beta = 1$, and we will initially consider the case when $\alpha = 1$. When $\alpha = 1$, Jensen's inequality tells us that generalists and specialists will be equally productive. Classically, this would suggest that specialists and generalists should have the same fitness.

And, indeed, all three cases have the same average v and the same average b (Â½ for each). However, the average fitnesses are all different.

Next, we consider the same three configurations, but with $\alpha = 0.9$. Jensen's inequality tells us that for this value of $\alpha$, generalists should have a higher average v and average b. Indeed, the average v and b is higher for generalists than for specialists: 0.536 versus 0.5. However, the average fitness of generalists, 0.287, is still lower than the average fitness of alternating specialists, 0.333.

These examples show that Jensen's inequality still holds, and still correctly tells us which configuration has the highest average v and average b. However, average v and average b are no longer directly proportional to average fitness. Therefore Jensen's inequality does not directly inform

average fitness, and we should not expect convex ROI functions to be required for specialists to be favored.

## Star graphs

Let $W_g$ be the fitness for the star shaped network group of generalists ($v = 0.5$, $b = 0.5$) and $W_s$ be the fitness of specialists (all of the points of the star get $v = 1$, $b = 0$ and the central point gets $v = 0$, $b = 1$).

Next assume there are $N$ cells on the points of the star and 1 cell in the center of the star. We then have:

$$W_s = \frac{N\beta}{2} \tag{8}$$

as the only individual with nonzero fitness is the central individual (all others have $b = 0$). The central individual's fecundity returns are $1^\alpha = 1$, and it's own viability returns are 0. However, the central individual gets shared $\frac{\beta}{2}$ of each of the $N$ other individuals' viability returns (which are $1^\alpha = 1$ each).

Next, for generalists, we have

$$W_g = N\left(\frac{1}{2}\right)^{2\alpha}\left(1 - \beta + \frac{\beta}{2} + \frac{\beta}{N}\right) + \left(\frac{1}{2}\right)^{2\alpha}\left(\frac{\beta}{N} + 1 - \beta + N\frac{\beta}{2}\right). \tag{9}$$

The term on the left of **Equation 9** comes from the fact that there are $N$ individuals each sharing $\frac{\beta}{2}$ of their viability returns (which is $\left(\frac{1}{2}\right)^\alpha$ each) with themselves, and are getting $\frac{\beta}{N}$ of the central individual's $\left(\frac{1}{2}\right)^\alpha$ viability returns shared with them. Additionally, they are getting $1 - \beta$ of their own viability returns (withheld from sharing). Finally, each of their fecundity returns is $\left(\frac{1}{2}\right)^\alpha$.

The term on the right of **Equation 9** represents the contribution to the group fitness of the single central individual. That individual gets $\frac{\beta}{2} * \left(\frac{1}{2}\right)^\alpha$ of viability returns shared to it $N$ times, and it also shares with itself and keeps a portion of its returns for itself. And it has a fecundity return of $\left(\frac{1}{2}\right)^\alpha$.

## Star topologies in the limit of large N

We first examine **Equation 9** in the limit where $N >> 1$:

$$W_g \approx N\left(\frac{1}{2}\right)^{2\alpha}\left(1 - \beta + \frac{\beta}{2}\right) + \left(\frac{1}{2}\right)^{2\alpha}\left(N\frac{\beta}{2}\right) \tag{10}$$

which reduces to

$$W_g \approx N\left(\frac{1}{2}\right)^{2\alpha}\left(1 - \beta + \frac{\beta}{2} + \frac{\beta}{2}\right) \tag{11}$$

and finally

$$W_g \approx N\left(\frac{1}{2}\right)^{2\alpha}. \tag{12}$$

To understand if generalists or specialists are favored we examine the ratio of generalist to specialist fitness $\frac{W_g}{W_s}$.

$$\frac{W_g}{W_s} = \frac{\beta}{2^{\alpha+1}}. \tag{13}$$

This means $W_g > W_s$ if $\beta > 2^{\alpha+1}$. Since $\beta$ and $\alpha$ are both bounded between 0 and 1, this is never achievable. Therefore, at large $N$, specialists are always favored.

## Star topologies in general

Alternately, if do not assume $N$ is large, but set $\beta = 1$ for simplicity, we can solve for the largest $\alpha$ which would yield equal fitness to generalists and specialists:

$$\alpha_{max} = -\frac{1}{2}\log_2\left(\frac{2N^2}{2N^2 + 2N + 8}\right). \tag{14}$$

## Rings with even and odd numbers of cells

In the main text we plot $\alpha^*$ for ring graphs with even numbers of cells. We made this choice as specialization is slightly frustrated when there are odd numbers of cells. Rings with odd numbers of cells must have at least one location at which like specialists are connected, thus slightly increasing the value of $\alpha^*$ compared to a ring with an even number of cells (*Appendix 1—figure 3*). However, as the size of the graph increases, a single frustrated pairing matters less and less, and the value of $\alpha^*$ for rings with odd numbers and large N appears to approach the value of $\alpha^*$ for rings with even numbers of cells.

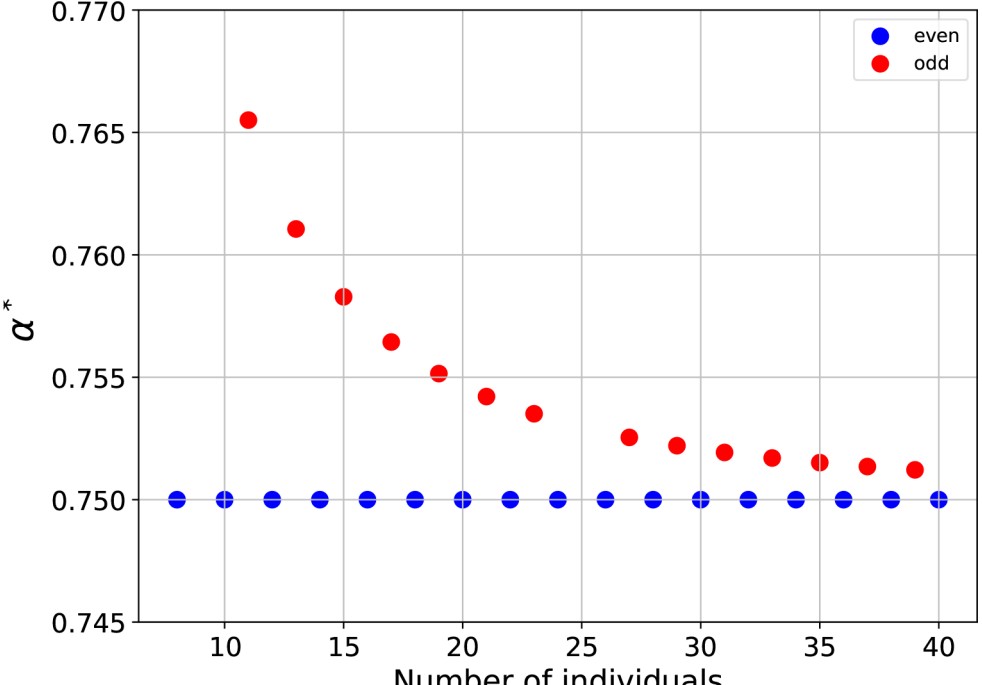

**Appendix 1—figure 3.** Rings with odd numbers of cells are frustrated. $\alpha^*$ plotted versus the number of cells in the ring, for rings with even and odd numbers of cells.

