## [Decision Letter]

**Acceptance summary:**

The evolution of germ-soma differentiation is one of the most fundamental questions in evolutionary biology, and the present paper investigates the consequences of altering one of the most basic assumptions: the traditional (symmetric) division of labor that has been studied from biology to economics. The authors consider a diversity of network structures and fitness functions and they find that sparser networks lead to higher levels of specialization.

**Decision letter after peer review:**

Thank you for submitting your article "Topological constraints in early multicellularity favor reproductive division of labor" for consideration by *eLife*. Your article has been reviewed by two peer reviewers, and the evaluation has been overseen by a Reviewing Editor and Diethard Tautz as the Senior Editor. The following individual involved in review of your submission has agreed to reveal their identity: Pierrick Bourrat (Reviewer #2).

The reviewers have discussed the reviews with one another and the Reviewing Editor has drafted this decision to help you prepare a revised submission.

Summary:

This is a very interesting paper on the evolution of germ-soma differentiation in which the authors consider the topology of interactions between the cells that make up the whole. They find that classical considerations about the convexity or concavity of certain functions characterizing the advantages of specialization no longer hold when the network topology is nontrivial. And such topologies are indeed found in nature. The reviewers were generally very supportive of the work but raised a number of points that need to be addressed in a revised manuscript.

Essential revisions:

1) The authors use a notion of fitness in which clonal cells can have different fitnesses, or more accurately, clonal groups can have different fitness. We know that there are some precedents in the literature, but this notion of fitness does not correspond to the notion of fitness one can associate with natural selection. To illustrate why, consider the following plant example. Take a single genet with two ramets in two environmental patches, one rich and one poor. Each ramet might adopt a very different developmental strategy from the other, considering the ecological constraints it is subjected to. These two strategies would nevertheless not be heritable in the sense that two offspring ramets put in the same environmental patch would develop the same developmental strategies (excluding noise). Thus, the differential success of each ramet is not an evolutionary success that can be associated with natural selection. This is a case analogous to the one presented by the authors. The notion of fitness they refer to seems to be rather the notion of realized fitness. This has no implications for the author's results per se but instead leads to an interpretation in which natural selection is not at work for explaining the division of labor in situations of concavity.

2) Related to the previous point, there seems to be a tension between, on the one hand, the claim that a concave function can lead to an increase in reproductive specialization, and on the other hand, claims that it has something to do with fitness. Fitness is about expected values, and in a situation of concave function, two or more cells specializing would yield a lower collective fitness than when not specializing. From a purely analytical point of view (i.e., Fisher's fundamental theorem), this seems impossible. So my question to the authors is whether there is not hidden somewhere a convex function, which is the relevant one for the evolutionary dynamics observed. Otherwise, what is the ecological explanation of such a result? There must be some ecological constraints that give rise to this phenomenon, and it would be good to know what the authors think they are.

3) There are well-known cases presented in the population genetics literature in which Fisher's fundamental theorem seems violated, but this is because of the environment (including the social environment) changes over time, such as frequency-dependent effects on an individual's success. We wonder if the results of the authors could not related to this literature in some way.

4) The model description is a bit abstract and occasionally hard to follow. It would be great to have fecundity and viability defined, and even better to have some real biological example of what returns on viability might mean and how they might be shared (I don't find the filamentous fungi example informative, at least not in the way it is written). That would also help the reader understand why there are returns on viability but not on fecundity. That the v_i_ vector is the "group investment strategy" also comes as a surprise and takes a bit for the reader to put it all together. Similarly, the existence of both a general adjancecy matrix and of a special case one that uses the *β*, is somewhat confusing the way it's described. If the authors anyway only work with the special case of equal sharing with the non-self neighbors then why not define the 1-*β*+*β*/ni quantities as c_ij_ when they appear in the text, and then write a fourth eqn for W in [1] that explicitly uses the *β*. That would certainly help the reader a lot.

5) Results subsection “Fixed resource sharing” first paragraph, we may be getting confused, but how can you vary *β* in the case when, as is now written, individual i "shares equally among interaction and self terms"? Doesn't this mean that *β* = 1?

6) “We conjecture that the troughs in Figure 3C, where specialization occurs for the lowest values of, occur when connectivity is just large enough so that a spanning tree is more likely to connect all individuals in the group than not”, we don't fully understand that conjecture: do the authors simply mean that the troughs occur when the random graph becomes connected with probability > 50%? (A spanning tree connects all individuals by definition.)

7) The authors suggest sparsity is the main determinant of whether a network supports reproductive specialization. But, their examples in 1B and 1C (where a ring is sparser than the bipartite network) to us suggest that it is not so much about "sparsity" as it is about "bipartiteness" – or how easy it is to subdivide the nodes into two classes such that most edges go between these two classes (that's what you'd want for specialization anyway, we guess), and that sparse graphs simply have a tendency to be close to bipartite. We suspect that a ring graph with an odd number of vertices will be less conducive to specialization (although you could still alternate germ/soma cells except at one point), and that a star graph where there is one node of degree n-1 and all the others have degree 1 may be an example of a sparse graph where evolving specialization is not so easy (because for this graph it's not clear how to divide the vertices into germ and soma).

8) Related to the previous point: we would be interested if the authors have considered what happens when the optimal strategy is not 1:1 but, say, 1:2. Does that make specialization more difficult? Here we think that, with a few additional simulations, the authors could add a lot to the paper in terms of the ability to connect properties of the graph (beyond comparing some explicit topologies and random graphs of varying sparsity) to its ability to support the evolution of reproductive specialization.

9) Finally, it would be nice to see how the different specialists are distributed on these networks (at least when the specialization is equal to 1). One can infer it, but we think it would visually help the reader to get the gist of how the model works very quickly.

---

## [Author Response]

Essential revisions:1) The authors use a notion of fitness in which clonal cells can have different fitnesses, or more accurately, clonal groups can have different fitness. We know that there are some precedents in the literature, but this notion of fitness does not correspond to the notion of fitness one can associate with natural selection. To illustrate why, consider the following plant example. Take a single genet with two ramets in two environmental patches, one rich and one poor. Each ramet might adopt a very different developmental strategy from the other, considering the ecological constraints it is subjected to. These two strategies would nevertheless not be heritable in the sense that two offspring ramets put in the same environmental patch would develop the same developmental strategies (excluding noise). Thus, the differential success of each ramet is not an evolutionary success that can be associated with natural selection. This is a case analogous to the one presented by the authors. The notion of fitness they refer to seems to be rather the notion of realized fitness. This has no implications for the author's results per se but instead leads to an interpretation in which natural selection is not at work for explaining the division of labor in situations of concavity.

Thank you for the thought-provoking comment. We are not completely sure if we understand your point correctly (and if not, we are happy to continue the discussion), but we are using the concept of fitness in the standard sense. While cells within groups can have different numbers of surviving offspring, in our simplified modeling world, selection does not act on these differences. Instead, we assume that the cells within each group are clonal, and that there can be genetic differences between groups. These genetic differences are responsible for cellular behavior, namely the extent to which they specialize in viability or fecundity tasks. Selection acts at the group level, in a way that is simply proportional to cellular productivity within these groups. Thus, selection acts between groups on differences in group fitness, which is caused by heritable variation in cellular behaviors underlying differentiation that vary between the groups.

However, we appreciate the reviewer’s point. We previously described mutations as affecting individual cells, which would of course mean that groups are no longer clonal. It is more accurate to say that in our simulation model, mutations change the pattern of specialization and sharing at different positions in the group, which we think of as being driven by a heritable developmental program. As with our analytical results, selection only acts on group-level fitness. When model parameters favor specialization, we see initially uniform groups in which every cell is a generalist gradually evolve developmental programs featuring complete reproductive specialization, providing a simulation test of our analytical results. We have thus revised and clarified the description of these simulations. We also stopped referring to the “fitness” of individual cells in the paper, since it is potentially confusing, instead describing the direct consequences of cellular specialization on their productivity (each cell’s 𝑣 ∗ 𝑏).

In addition to the changes noted in the paragraph above, we modified the model description to read:

“We consider a model of multicellular groups composed of clonal cells that each invest resources into viability and fecundity. Because there is no within-group genetic variation, within-group evolution is not possible, though selection can act on group-level fitness differences. Specifically, we consider the pattern of cellular investment in fecundity and viability, and their sharing of these resources with neighboring cells within the group, to be the result of a heritable developmental program. Thus, selection is able to act on the multicellular fitness consequences of different patterns of cellular behavior within the group.”

We modified the simulation description to read:

“These simulations begin with no resource sharing (i.e., 𝛽 = 0); during every round, each group in the population has a 2% chance that a mutation will impact its developmental program, and the 𝛽 value for one of its cells will change.”

2) Related to the previous point, there seems to be a tension between, on the one hand, the claim that a concave function can lead to an increase in reproductive specialization, and on the other hand, claims that it has something to do with fitness. Fitness is about expected values, and in a situation of concave function, two or more cells specializing would yield a lower collective fitness than when not specializing. From a purely analytical point of view (i.e., Fisher's fundamental theorem), this seems impossible. So my question to the authors is whether there is not hidden somewhere a convex function, which is the relevant one for the evolutionary dynamics observed. Otherwise, what is the ecological explanation of such a result? There must be some ecological constraints that give rise to this phenomenon, and it would be good to know what the authors think they are.

Thank you again for this question. We believe the reviewers have found a very important point that lacked clarity in the previous version of our manuscript. We appreciate the deep question regarding the presence of a hidden convexity. To be frank, we were surprised by this result at first as well. However, this surprise stems from the fact that the mathematical rule connecting specialization and the second derivative of the return on investment (ROI) function does not apply to asymmetric trade on sparse networks.

The following response explains the logic underlying this result and shows why there are no hidden convexities in our model. We apologize for its length, but since it is central to the paper we wanted to make the math absolutely clear.

In our model we make the simple assumption that group fitness is directly proportional to the sum of cellular productivity (each cell’s 𝑣 ∗ 𝑏) within the group. We see this as having the simplest biological interpretation, groups that generate a larger number of cells can make more progeny (i.e., more groups), all else equal. As a result, while holding the number of cells per group constant, the group with the highest fitness will be composed of cells with the highest average fitness.

To understand why a hidden convexity need not be present here, we will first discuss the mathematical rule that typically connects specialization and the second derivative of the ROI function: Jensen’s inequality. Jensen’s inequality states that for a convex function F(x), the average value of 𝐹(𝑥), ⟨𝐹(𝑥)⟩, is larger than 𝐹(⟨𝑥⟩), where ⟨𝑥⟩ is the average value of 𝑥. In other words, ⟨𝐹(𝑥)⟩ > 𝐹(⟨𝑥⟩). A corollary of Jensen’s inequality is that the opposite is true for concave functions, i.e., for a concave function 𝐺(𝑥), ⟨𝐺(𝑥)⟩ < 𝐺(⟨𝑥⟩). We assume this is the rule the reviewers refer to when they say “in a situation of concave function, two or more cells specializing would yield a lower collective fitness than when not specializing.”

For the traditional case of fully connected topologies and symmetric sharing, Jensen’s inequality provides a mathematical connection between a group’s fitness and its degree of specialization. With fully connected topologies, each cell shares its returns on 𝑣 and 𝑏 equally with all cells. As a result, all cells end up with 𝑣 equal to the average 𝑣 and 𝑏 equal to the average 𝑏. As a result, each cell’s 𝑣 ∗ 𝑏 is thus the average of 𝑣 multiplied by the average of 𝑏. Thus, group fitness is directly proportional to average 𝑣 and average 𝑏. Jensen’s inequality guarantees that for convex ROI functions, the average v produced by specialists (⟨𝐹(𝑥)⟩) is higher than the average 𝑣 produced by generalists (𝐹(⟨𝑥⟩)). The same is true for 𝑏. Since specialists must have larger average 𝑣 and 𝑏 than generalists, group fitness must be higher as well. The same argument holds in reverse for concave functions, i.e., if the ROI function is concave, the average 𝑣 and average 𝑏 are lower for specialists than for generalists so group fitness is lower, too. Thus, for fully connected topologies and symmetric sharing, Jensen’s inequality allows one to connect group fitness to average 𝑣 and average 𝑏.

Crucially, however, the connection between ROI convexity/concavity and fitness is indirect. Jensen’s inequality directly relates the degree of specialization to the average 𝑣 and average 𝑏; Jensen’s inequality does not itself say anything about group fitness. Group fitness is proportional to the average of the product 𝑣 ∗ 𝑏, which does not have to be directly proportional to average 𝑣 or average 𝑏 (even though for fully connected topologies and symmetric sharing ⟨𝑣𝑏⟩ is directly proportional to ⟨𝑣⟩ and ⟨𝑏⟩).

That Jensen’s inequality connects the degree of specialization to average 𝑣 and average 𝑏, but not group fitness is a crucial distinction. Jensen’s inequality is a mathematical truism, and cannot be violated. Indeed, for any concave ROI function, generalists will produce more 𝑣 and 𝑏 than specialists. This fact is just as true for sparse topologies and asymmetric sharing as it is for fully connected topologies and symmetric sharing. However, for sparse topologies and asymmetric sharing, group fitness is not directly proportional to average 𝑣 and average 𝑏; instead, group fitness strongly depends on network structure as well.

To understand how group fitness decouples from average 𝑣 and average 𝑏 for sparse topologies and asymmetric sharing, consider a ring of four cells in three different configurations: one that alternates between viability and fecundity specialists, one in which like-specialists are connected to each other, and one in which all cells are generalists (pictured in Author response image 1; red cells are viability specialists, blue cells are fecundity specialists, and purple cells are generalists). For simplicity, we will set 𝛽 = 1, and we will initially consider the case when 𝛼 = 1. When 𝛼 = 1, Jensen’s inequality tells us that generalists and specialists will be equally productive. Classically, this would suggest that specialists and generalists should have the same fitness.

**Author response image 1. sa2fig1:** 

And, indeed, all three cases have the same average 𝑣 and the same average 𝑏 (½ for each). However, the group fitnesses are all different. Note, depending on the configuration, ⟨𝑣⟩⟨𝑏⟩ can be greater than, less than, or equal to ⟨𝑣𝑏⟩.Next, we consider the same three configurations, but with 𝛼 = 0.9. Jensen’s inequality tells us that for this value of 𝛼, generalists should have a higher average 𝑣 and average 𝑏. Indeed, the average 𝑣 and 𝑏 is higher for generalists than for specialists: 0.536 versus 0.5. However, the group fitness of generalists, 1.15, is still lower than the group fitness of alternating specialists (i.e., leftmost configuration in the schematic in Author response image 1), 1.333.

These examples show that Jensen’s inequality still holds, and still correctly tells us which configuration has the highest average 𝑣 and average 𝑏. However, average 𝑣 and average 𝑏 are no longer directly proportional to group fitness. Therefore, Jensen’s inequality does not directly inform group fitness, and we should not expect convex ROI functions to be required for specialists to be favored.

Ultimately, the difference between the classic fully connected topologies and the asymmetric sharing / sparse topologies we consider is the ability of viability specialists to preferentially share viability to fecundity specialists. In fully connected topologies, some of the potential benefit from specialization is wasted. Consider the case where viability specialists share viability with other viability specialists: this is entirely unhelpful, two viability specialists (with 0 fecundity) that help each other survive still both have zero fitness. When the return on investment function is linear (and the graph is fully connected), the effect of sharing with the same cell type (as opposed to a complementary cell type) exactly cancels out the benefits of specializing; trade is only beneficial once the return on investment function becomes convex. In this case, specialists make enough extra *v* and *b* that groups of specialists do better than groups of generalists.

Again, these results were surprising. To help establish intuition for the role of preferentially connecting unlike specialists, we developed a mean field model. Based on the average number of connections per cell, this model determines the fraction of connections that must be between unlike specialists for the network to support specialization with concave ROI. We find that if unlike specialists are preferentially connected, specialization despite concave ROI should be expected for a wide range of networks. These results are similar to our simulations of randomly generated graphs (see Figure 3 in the main text), in which we also observed specialization for a wide range of parameters. Combined, these observations suggest that specialization, despite concave ROI, does not require precisely designed topologies, but is a general principle applicable to many different network structures. We agree that this was not clear in the original manuscript, and have added this model to the main text as described below.

We added Discussion paragraphs on Jensen’s inequality:

“To understand how specialization can be favored despite concave return on investment (ROI) curves, consider Jensen’s inequality. […] With asymmetric sharing and sparse topologies, Jensen’s inequality still informs the average viability and fecundity produced, but does not directly inform the group fitness.”

We also added a mean field model to help the reader develop intuition for these results:

“Mean field model

Finally, to capture some general principles underlying this phenomenon, we consider a mean-field model with <inline-graphic mime-subtype="png" mimetype="image" xlink:href="media/image1.png" /> cells, each of which is connected to <inline-graphic mime-subtype="png" mimetype="image" xlink:href="media/image2.png" /> other cells. […]However, this inequality presents a useful heuristic which can be used to determine if specialization is favored by measuring just a few properties of the graph.”

We also added a Discussion paragraph on directed/wasted effort:

“Finally, we note that the primary benefit of sparsity is that sparse networks are likely to be at least somewhat bipartite. The more bipartite-like a network is, the less effort is wasted, and the easier it is for specialization to be favored.”

Finally we added a section on the four cell network:

“Jensen's inequality and sparse topologies and asymmetric sharing

To understand how average fitness decouples from average 𝑣 and average 𝑏 for sparse topologies and asymmetric sharing, consider a ring of four cells in three different configurations: one that alternates between viability and fecundity specialists, one in which like-specialists are connected to each other, and one in which all cells are generalists. […] Therefore, Jensen’s inequality does not directly inform average fitness, and we should not expect convex ROI functions to be required for specialists to be favored.”

3) There are well-known cases presented in the population genetics literature in which Fisher's fundamental theorem seems violated, but this is because of the environment (including the social environment) changes over time, such as frequency-dependent effects on an individual's success. We wonder if the results of the authors could not related to this literature in some way.

This question seems to be related to the central issue raised above – is there a hidden feature in the model that makes specialization beneficial *despite* a concave investment function (i.e., by adding convexity somewhere or introducing some frequency-dependent or environmental effect)? Above, we explained the logic of why specialization can be favored despite concave investment functions.

Also, to be clear, we don’t see our work as violating Fisher’s fundamental theorem, which states that selection will increase the mean fitness of a population at a rate proportional to the genetic variation in fitness in the population. Because our groups are clonal, all the genetic variation in our population occurs between groups, not within groups, and selection is acting only on group-level fitness. Thus, Fisher’s fundamental theorem is at a limit within groups (as there is no within-group genetic variation, there is no within-group evolution).

4) The model description is a bit abstract and occasionally hard to follow. It would be great to have fecundity and viability defined, and even better to have some real biological example of what returns on viability might mean and how they might be shared (I don't find the filamentous fungi example informative, at least not in the way it is written). That would also help the reader understand why there are returns on viability but not on fecundity. That the v_i_ vector is the "group investment strategy" also comes as a surprise and takes a bit for the reader to put it all together. Similarly, the existence of both a general adjancecy matrix and of a special case one that uses the *β*, is somewhat confusing the way it's described. If the authors anyway only work with the special case of equal sharing with the non-self neighbors then why not define the 1-*β*+*β*/ni quantities as c_ij_ when they appear in the text, and then write a fourth eqn for W in [1] that explicitly uses the *β*. That would certainly help the reader a lot.

These are great points, and we agree that it is crucial that the model and its biological meaning are clear to the reader.

The dichotomy of viability vs. fecundity was originally used by Michod to partition components of cellular fitness into actions that contribute to keeping a cell alive (viability), and actions that directly contribute to reproduction (fecundity). The intuition underlying this is that multicellular organisms often have evolved to divide labor along these two lines (i.e., reproduction of the organism by germ cells and survival provided by somatic cells), while their unicellular ancestors had to do both. We define viability as an activity (Michod uses the term “effort”) that keeps the cell alive (e.g., investing in cellular homeostasis or behaviors that improve survival), and fecundity more narrowly as effort involved in cellular reproduction itself. At the cellular level, there appears to be a fundamental asymmetry in how viability effort and fecundity effort can be shared among cells: while multicellular organisms readily evolve differentiated cells that are completely reliant on helper cells for survival (i.e., glial cells that support neurons in animals or companion cells that support sieve tube cells in plants), no cell can directly share its ability to reproduce. To better understand the intuition behind this, consider a cell that elongates prior to fission. This cell must grow to approximately twice its original length. Two cells cannot elongate by 50% and then combine their efforts; elongation is an intrinsically single cell effort.

While we believe this was also Michod’s interpretation of “reproductive effort”, this concept, like many in biology, is subject to interpretation. While it is clear the cellular behaviors underlying replication cannot be shared, resources required for reproduction could have been provided by another cell. Fortunately, our main conclusions are general and can accommodate various definitions of reproductive effort, which include sharing. While we present the simplest case (no sharing of reproductive effort) to help explain our paper’s central idea as clearly as possible, we do not need to forbid sharing of reproductive effort in order for specialization to evolve with concave ROI functions. As long as there is a significant asymmetry in how much 𝑣 and 𝑏 are shared, specialization with concave ROI functions will evolve. We explore this generalization in Figure 2. Because much of the effort involved in reproduction is, by its nature, unshareable, we believe that asymmetry in sharing between 𝑣 and 𝑏 is a general feature of multicellular systems. Of course, our results are far more general than the evolution of multicellularity, and should apply to many systems in which entities trade along networks (biological, economic, etc.). We have updated the description in the text of how cells can share viability, illustrating the general point with additional examples.

Finally, we have updated our description of the model to clarify the roles of 𝛽 and 𝑐_𝑖𝑗_.

We modified the “Model” section to read:

“Reproductive specialization can be modeled as the separation of two key fitness parameters, those related to either viability or fecundity, into separate cells within the multicellular organism (13,35). […]Thus, selection is able to act on the multicellular fitness consequences of different patterns of cellular behavior within the group.”

And added clarifying statements like:

“In other words, 𝑐_𝑖𝑖_ = 1 − 𝛽, 𝑐_𝑖𝑗_ = 𝛽/𝑛_𝑖_ if cells 𝑖 and 𝑗 are connected, and 𝑐_𝑖𝑗_ = 0 if cells 𝑖 and 𝑗 are not connected.”

5) Results subsection “Fixed resource sharing” first paragraph, we may be getting confused, but how can you vary *β* in the case when, as is now written, individual i "shares equally among interaction and self terms"? Doesn't this mean that *β* = 1?

We do not allow cells to give all of their viability returns away, as this would result in cells that are not viable. Cells keep (1 − 𝛽) of their viability returns, and designate *β* of their viability returns for sharing. They then split their viability returns designated for sharing into 𝑁 + 1 equally sized portions, give one portion to each of their 𝑁 neighbors, and keep one portion for themselves. In other words, the *β* portion that is designated “for sharing” is not all given away, but is split equally among the connected cells and the cell itself. We have updated the explanation accordingly.

We removed the sentence “Individual 𝑖 shares 𝑣_𝑖_^𝛼^ equally among interaction and self terms.” that was on line 146. The model was explained in the previous section, so this statement was ultimately redundant and unclear. Instead, we modified the model explanation to read:

“Cells cannot give away all of their viability returns, as they would no longer be viable; mathematically, we count a cell among its neighbors and thus ensure that they always “share” a positive portion of viability returns with themselves, so that 𝑐_𝑖𝑖_ > 0.”

And

“…and when 𝛽 = 1 cells share everything equally among all connections and themselves.”

6) “We conjecture that the troughs in Figure 3C, where specialization occurs for the lowest values of, occur when connectivity is just large enough so that a spanning tree is more likely to connect all individuals in the group than not”, we don't fully understand that conjecture: do the authors simply mean that the troughs occur when the random graph becomes connected with probability > 50%? (A spanning tree connects all individuals by definition.)

The reviewers are correct that a spanning tree connects all individuals. We were attempting to state this fact using technical language that would be clear to experts (50% probability that a spanning tree was present) as well as with more general language accessible to non-experts (50% probability that all individuals connected). We have updated this statement to be clear to all.

We modified this statement to read:

“We conjecture that the troughs in Figure 3B, where specialization occurs for the lowest values of 𝛼, occur when connectivity is just large enough so that the existence of a spanning tree is more likely than not.”

7) The authors suggest sparsity is the main determinant of whether a network supports reproductive specialization. But, their examples in 1B and 1C (where a ring is sparser than the bipartite network) to us suggest that it is not so much about "sparsity" as it is about "bipartiteness" – or how easy it is to subdivide the nodes into two classes such that most edges go between these two classes (that's what you'd want for specialization anyway, we guess), and that sparse graphs simply have a tendency to be close to bipartite. We suspect that a ring graph with an odd number of vertices will be less conducive to specialization (although you could still alternate germ/soma cells except at one point), and that a star graph where there is one node of degree n-1 and all the others have degree 1 may be an example of a sparse graph where evolving specialization is not so easy (because for this graph it's not clear how to divide the vertices into germ and soma).

Thank you for this point, with which we largely agree; “bipartiteness” is more important than sparsity. Sparsity is important for two reasons. First, as the reviewers suggest, sparse graphs tend to be close to bipartite. Second, sparse topologies are common in nature. Thus, sparsity may be a common natural route to being somewhat close to bipartite. We have modified the text to make this point clear. However, as will be highlighted by our discussion below, additional features of the graph, such as connectivity, can play a role in determining if specialization is favored as well.

We also appreciate the star graph suggestion, which is a very interesting topology. This graph is also relevant to understanding the evolution of asymmetric investment in germ and somatic cells, as is common in among independently-evolved multicellular organisms, like animals, plants, and volvocine green algae. Hopefully after clarifying our model in response to previous questions it is now clear that the star graph can strongly favor specialization. We now work this example in detail in the supplement. Briefly, we will discuss a five cell star here.

**Author response image 2. sa2fig2:** Cartoon image generalists (a) and specialists (b) in a five cell star graph topology. Wright-Fisher simulations of five cell star topologies for a range of 𝛼 and 𝛽 values.

If the central cell were a viability specialist, specialists would have a lower fitness than generalists. However, if the central cell is a fecundity specialist, then the four surrounding viability specialists provide the central cell with enough viability to make this configuration favored. In fact, for large groups (𝑁 > > 1), specialists are always favored – for any 𝛽 and any 𝛼. For the five cell examples shown above, specialists are favored for *α* values as low as 0.222; a heat map relating degree of specialization to 𝛼 and 𝛽 is shown in Author response image 3.

Finally, the reviewers are correct about rings with even and odd numbers of cells. The effect is small, but it is true that specialization is frustrated in rings with odd numbers of cells. In fact, we cut a discussion of this phenomenon from our manuscript as we worried that it would be a distraction. However, it is now clear that its absence in fact raises questions, so we have restored it.

**Author response image 3. sa2fig3:** 

Here, 𝛼^∗^, the lowest value of 𝛼 for which specialization is favored, is plotted versus the number of cells in the ring, for rings with even and odd numbers of cells.As discussed above, we added a Discussion paragraph that reads:

“Finally, we note that the primary benefit of sparsity is that sparse networks are likely to be at least somewhat bipartite. The more bipartite-like a network is, the less effort is wasted, and the easier it is for specialization to be favored.”

We added a section analytically working out when specialists and generalists are favored for star topologies with various N, 𝛼, and 𝛽.

We added a section discussing even and odd numbered rings.

8) Related to the previous point: we would be interested if the authors have considered what happens when the optimal strategy is not 1:1 but, say, 1:2. Does that make specialization more difficult? Here we think that, with a few additional simulations, the authors could add a lot to the paper in terms of the ability to connect properties of the graph (beyond comparing some explicit topologies and random graphs of varying sparsity) to its ability to support the evolution of reproductive specialization.

Thank you for this interesting suggestion. First, we return to the mean field model introduced in comment 2, but now allow the fraction of fecundity specialists to be 𝑋 (rather than forcing 𝑋 = ½). The model itself is presented in detail in the text changes below; here, we summarize the results.

Our mean field model suggests that a larger proportion of fecundity specialists makes concave specialization easier to achieve. Further, we find that concave specialization is only possible if more than one fourth of cells are fecundity specialists.

We stress here that this is a mean field model, and does not apply to scenarios like the star network, in which cells have very different individual values of 𝑧. If such networks do or do not favor specialization for 𝛼 < 1 will again be a graph coloring problem.

We added a section on varying ratios of specialists:

“Effect of varying ratios of specialists

We now allow the fraction of fecundity specialists to be 𝑋 (rather than forcing 𝑋 = 1/2). […] If such networks do or do not favor specialization for 𝛼 < 1 will again be a graph coloring problem.”

9) Finally, it would be nice to see how the different specialists are distributed on these networks (at least when the specialization is equal to 1). One can infer it, but we think it would visually help the reader to get the gist of how the model works very quickly.

We have visualized investment in different tasks on fully connected, ring, and bipartite graphs. We agree that these images are instructive for the reader.

In Author response image 4 is an image of specialists in a nearest neighbor topology:

**Author response image 4. sa2fig4:** 

In Author response image 5 for the bipartite network, complete specialization happens readily:

**Author response image 5. sa2fig5:** 

Author response image 6 for the complete network, generalists dominate with some fluctuations:

**Author response image 6. sa2fig6:** 

We modified Figure 1 to incorporate these images:

**Author response image 7. sa2fig7:** Evolution of resource sharing. (**a**) Initially, individuals do not share resources; however, they may evolve to do so via random mutations. Here, the mean specialization of the fittest of 100 groups each with 10 cells after 100,000 steps is plotted as a function of specialization power. Error bars are standard deviations across 10 replicates. Blue is the fully connected network, red is the neighbor network, and green is the balanced bipartite topology. (**b-d**) The final distribution of specialization values for individual cells in fully connected (**b**), nearest-neighbor (**c**), and balanced bipartite topologies (**d**). The color of cells in b-d represents their degree of specialization, as indicated in the scale bar.